# Unwinding of a eukaryotic origin of replication visualized by cryo-EM

**Sarah S. Henrikus** [1,4], **Marta H. Gross**[2], **Oliver Willhoft**[1], **Thomas Pühringer**[1], **Jacob S. Lewis** [1], **Allison W. McClure**[2,5], **Julia F. Greiwe** [1], **Giacomo Palm**[1], **Andrea Nans** [3], **John F. X. Diffley** [2] & **Alessandro Costa** [1] ✉

To prevent detrimental chromosome re-replication, DNA loading of a double hexamer of the minichromosome maintenance (MCM) replicative helicase is temporally separated from DNA unwinding. Upon S-phase transition in yeast, DNA unwinding is achieved in two steps: limited opening of the double helix and topological separation of the two DNA strands. First, Cdc45, GINS and Polε engage MCM to assemble a double CMGE with two partially separated hexamers that nucleate DNA melting. In the second step, triggered by Mcm10, two CMGEs separate completely, eject the lagging-strand template and cross paths. To understand Mcm10 during helicase activation, we used biochemical reconstitution with cryogenic electron microscopy. We found that Mcm10 splits the double CMGE by engaging the N-terminal homo-dimerization face of MCM. To eject the lagging strand, DNA unwinding is started from the N-terminal side of MCM while the hexamer channel becomes too narrow to harbor duplex DNA.

All known helicases involved in DNA replication assemble as hexameric rings that unwind the double helix by threading one DNA strand through their central channel. Different helicases access the single-stranded translocation substrate through mechanisms that evolved independently, but only some of these are understood at the molecular level. For example, rolling-circle replication of certain bacteriophages, viruses and plasmids involves DNA nicking to create a single-stranded DNA (ssDNA) end through which the helicase ring is threaded[1]. In bacterial chromosome replication, instead, DNA is melted so that the replicative helicase can be loaded around ssDNA[2]. How eukaryotes perform this task is unclear. It is known that the minichromosome maintenance (MCM) replicative helicase is loaded around duplex DNA, and, upon entry into S phase, it switches to encircling ssDNA, but the molecular basis for such transition remains unknown. Understanding this process is critical because its regulation ensures that any DNA segment in the genome is copied only once per cell cycle, preventing the rise of chromosome instability and the onset of cancer[3,4].

The MCM helicase is formed of six homologous ATPase subunits[3]. During late mitosis and throughout G1 phase, two MCM rings are loaded around duplex DNA at origins of replication, forming a double hexamer with N-terminal zinc finger (ZnF) domains engaged in dimerization. Although double hexamers provide the symmetry to support bidirectional replication, they are catalytically inactive[5–9], and cells need to enter S phase before the helicase function is switched on. In vitro reconstitution efforts established that activation of DNA unwinding can be staged. In the first step, DNA melting is nucleated, and, in the second step, the two DNA strands become topologically segregated[10]. Biochemical studies led to identifying the factors needed for the first nucleation step, and cryogenic electron microscopy (cryo-EM) imaging of the holo-helicase assembled around the double helix explained the structural mechanism for breaking base pairing. Here, a set of firing factors recruit Cdc45, GINS and Polε to the double hexamer forming a double Cdc45-MCM-GINS-Polε (dCMGE) structure. In this complex, 0.7 turns of the double helix become untwisted inside each MCM, and at least three base pairs are broken. To accommodate untwisting,

[1]Macromolecular Machines Laboratory, Francis Crick Institute, London, UK. [2]Chromosome Replication Laboratory, Francis Crick Institute, London, UK. [3]Structural Biology Science Technology Platform, Francis Crick Institute, London, UK. [4]Present address: Genome Stability Unit, St. Vincent's Institute of Medical Research, Fitzroy, Victoria, Australia. [5]Present address: Department of Biochemistry and Molecular Genetics, University of Colorado Anschutz Medical Campus, Aurora, CO, USA. ✉e-mail: alessandro.costa@crick.ac.uk

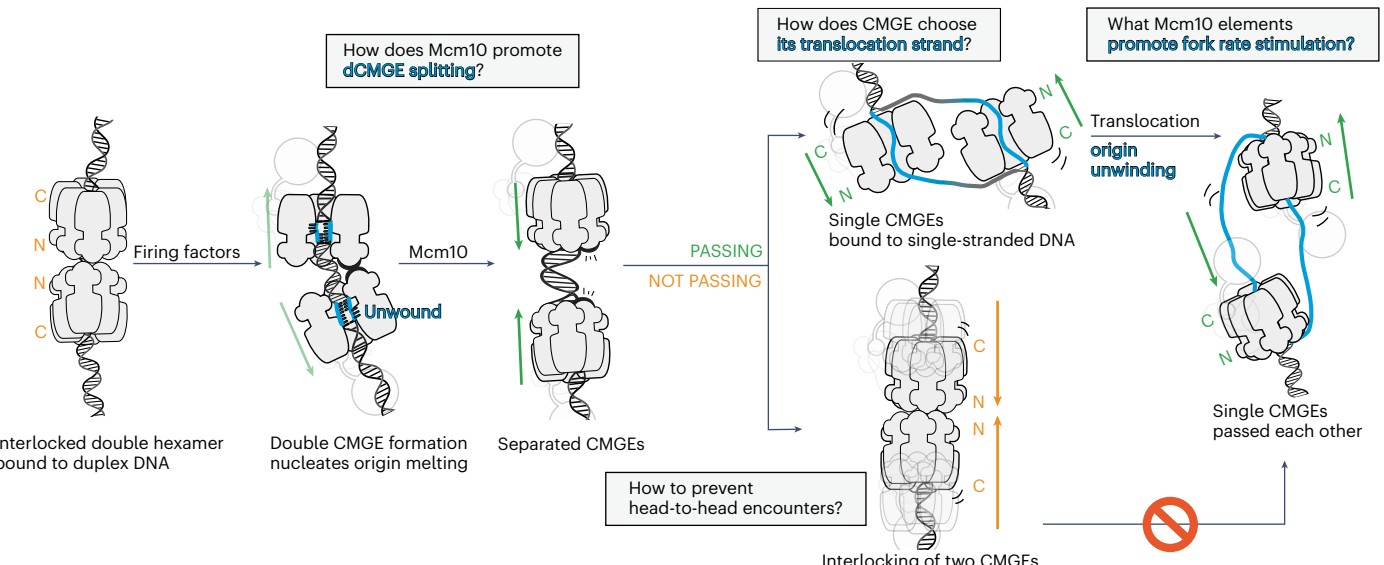

**Fig. 1 | Cartoon representation of helicase activation at a eukaryotic origin of replication.** MCM is loaded onto duplex DNA as a double hexamer with dimerizing N-termini. Upon S-phase transition, Cdc45, GINS and Polε are recruited to MCM, forming a dCMGE, causing partial separation of two helicases, exposure of duplex DNA in between the two helicase rings and nucleation of DNA melting within each ATPase core. Mcm10 recruitment splits the dCMGE and causes two single helicases to move toward one another and cross paths. How this happens is unclear. In addition, how the CMG discriminates between the strand to be ejected and the strand to be retained in the ATPase pore needs to be defined. Finally, the fork rate stimulation function by Mcm10 and its modulation through phosphorylation by Rad53 are poorly understood.

the two helicases move away from one another, remaining tethered via Mcm6 alone, while duplex DNA becomes exposed between the two MCM rings[10,11]. The second activation step is then triggered by the recruitment of Mcm10, a single-stranded binding protein[12] that contains N-terminal OB-fold and ZnF domains and a poorly characterized C-terminal region. Mcm10 latches across MCM, stretching from N-terminus to C-terminus[13–15], and induces a complete rearrangement of the origin of replication. The dCMGE separates into two single CMGEs (sCMGEs) that start translocating toward one another, with N-terminal to N-terminal directionality. However, the MCMs do not re-engage in dimer formation; rather, the sCMGEs cross paths so that what was the N-terminal dimerization domain in the double hexamer becomes the leading edge of the advancing helicase. Learning what splits the two MCM hexamers and promotes helicase crossing is important to understand replication initiation.

For two sCMGEs to cross paths, each helicase must transition from duplex to ssDNA binding[16]. To achieve this, 1.5 turns of the double helix entrapped in each MCM hexamer, as observed in the dCMGE structure, must be unwound, and one strand must be ejected from the central MCM channel. Unwinding is known to be achieved through the Mcm10-dependent activation of the ATPase function of the MCM hexamer, but how this physically happens is unknown (Fig. 1). Is DNA unwound from the N-terminal or the C-terminal side of MCM first? Moreover, how does MCM determine which strand to retain inside the ring pore and which strand to eject? Is the single-stranded binding function of Mcm10 required in this process? Addressing these issues is important to establish how sCMGE gains access to the single-stranded translocation substrate to topologically separate the two DNA strands.

Mcm10 contains a poorly characterized fork rate stimulation activity that can be selectively inhibited when DNA damage is detected, through phosphorylation by checkpoint kinase Rad53. A separate fork rate stimulation role is also provided by Mrc1, a member of the fork stabilization complex together with Csm3 and Tof1, understood to act redundantly with Mcm10 (ref. 17). What Mcm10 elements modulate fork stimulation and how Mcm10 and Mrc1 functionally interact remain unclear. To understand the essential role of Mcm10 and its regulation during initiation and elongation, we used biochemical reconstitution and cryo-EM to dissect the origin activation reaction.

## Results

### Mcm10 recruitment splits the CMGE dimer

To reconstitute Mcm10-dependent origin DNA unwinding with purified proteins (Extended Data Fig. 1a), we loaded double hexamers onto ARS1 origin DNA. The double helix was capped at both leading strand ends with a covalently linked HpaII methyltranferase (MH) roadblock to stop the helicase from sliding off DNA[9,18] (Extended Data Fig. 1b). After double hexamer phosphorylation by DDK, purified phospho-double-hexamer-DNA complexes were incubated with firing factors required for CMGE formation (CDK, Sld2, Sld3/7, Dpb11, Polε, GINS and Cdc45). We omitted or included Mcm10 to induce, respectively, nucleation of DNA melting or full origin DNA unwinding (Fig. 2a). Negative-stain electron microscopy analysis revealed that 32 ± 10% of double hexamers were converted to dCMGE when Mcm10 was omitted, consistent with our previous work[11]. When Mcm10 was included, 51 ± 15% of MCM-containing particles were identified as sCMGE (corresponding to a 34 ± 13% double-hexamer-to-sCMGE conversion efficiency), whereas no dCMGE was detected (Fig. 2b and Extended Data Fig. 1c,d). A staged reaction, where dCMGE was assembled before the addition of Mcm10, resulted in dCMGE-to-sCMGE conversion (although with slightly lower double-hexamer-to-sCMGE conversion efficiency; Extended Data Fig. 1e,f). We conclude from these results that Mcm10 splits a dimer of CMGE into two monomeric CMGE helicases, compatible with earlier observations that single CMGs assembled at origins of replication cross paths when DNA unwinding is activated[10].

### Mcm10 binds MCM homo-dimerization interface

To understand the mechanism of dCMGE splitting by Mcm10, we inspected the sCMGE particles. We identified an unassigned density feature decorating the N-terminal face of MCM exposed upon dCMGE splitting (Fig. 2b). We reasoned that this N-terminal-MCM-interacting factor could be one of two entities: (1) the MH roadblock marking the end of the origin DNA substrate that could be reached after sCMGEs cross paths or (2) the Mcm10 firing factor itself. To discriminate

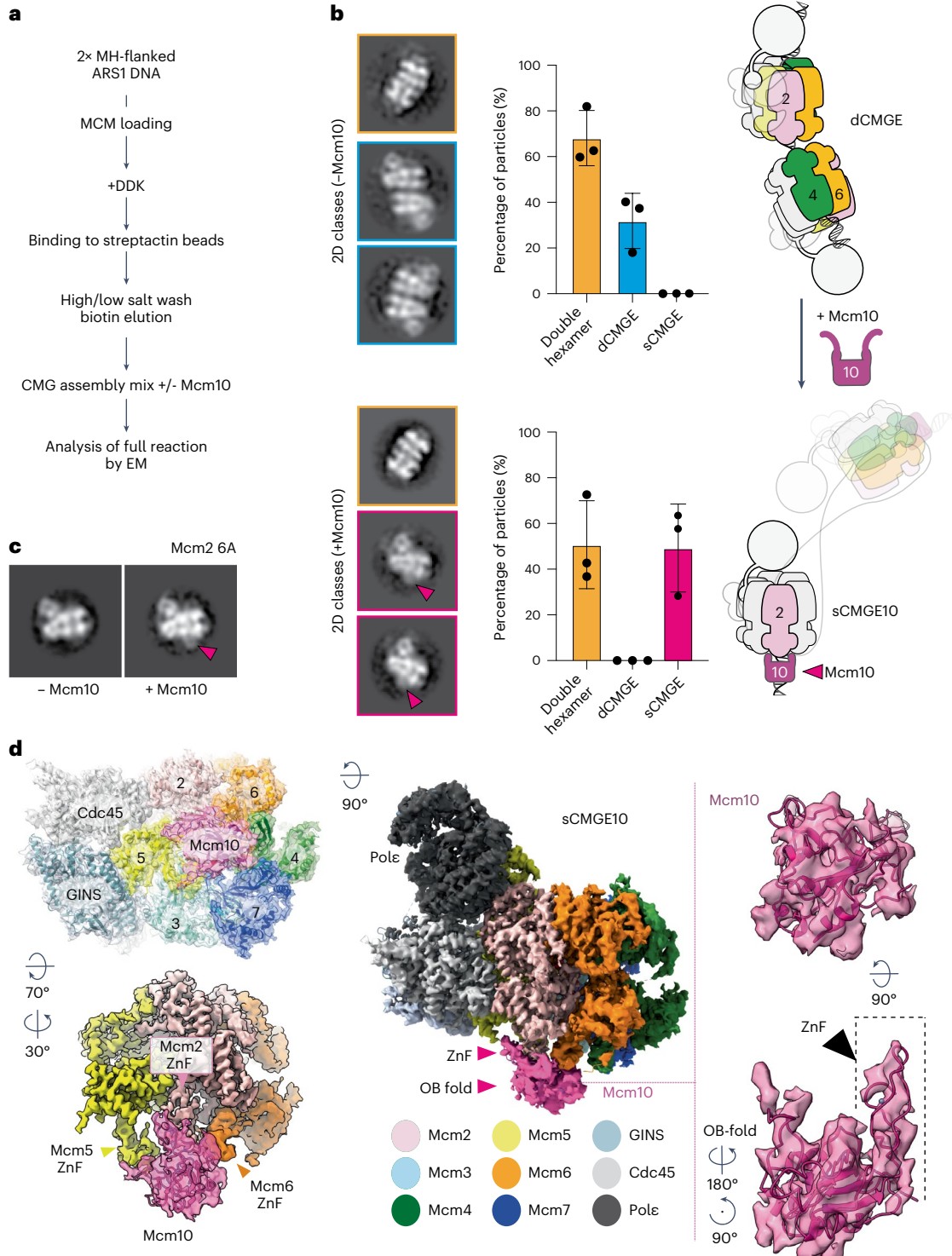

**Fig. 2 | Mcm10 binds N-terminal MCM and splits dCMGE. a**, Workflow for single CMGE10 assembly at a roadblocked origin of replication. **b**, CMGE assembly reaction in the absence of Mcm10 yields dCMGEs, whereas sCMGEs are formed when Mcm10 is added. sCMGEs appear decorated at the N-terminal domain. Left panel contains 2D averages; middle panel shows particle quantitation; and right panel depicts dCMGE splitting by Mcm10. The experiment was performed three times. Error bars, mean ± s.d. **c**, sCMGE10 formation with the

Mcm2 6A variant that forms sCMGEs in the absence of Mcm10 establishes that the N-terminal density is Mcm10. **d**, Composite map showing sCMGE (apix = 1.08 sigma = 0.00828) and additional density observed after particle binning and Pol epsilon signal subtraction (apix = 2.16, sigma = 0.0157). Atomic modeling indicates that the density decorating the N-terminal MCM corresponds to the OB-fold and ZnF domains of Mcm10.

between these two possibilities, we used an MCM variant with changes in the C-terminal ATPase motor (Mcm2 6A), which can make CMGE to wild-type levels but forms non-decorated sCMGE even in the absence of Mcm10 and fails to unwind DNA in origin activation reactions[11].

When sCMGE^Mcm2 6A was assembled without Mcm10, decoration was still not visible (Extended Data Fig. 1g). However, N-terminal MCM decoration was observed when Mcm10 was added to the reaction (Fig. 2c and Extended Data Fig. 1h). As Mcm2 6A cannot unwind DNA even in

the presence of Mcm10 (ref. [11]) and is static on duplex DNA according to single-molecule imaging measurements[19], the MH roadblock is not a plausible candidate for the N-terminal density. This is because reaching MH would require that sCMGEs translocate along the unwound ssDNA and cross paths. Thus, the feature decorating the N-terminal tier of MCM in sCMGE is, most likely, Mcm10.

To confirm our assignment, we used cryo-EM to image the origin unwinding reaction with wild-type MCM and Mcm10. We determined a first consensus sCMGE structure to 3.7-Å resolution based on approximately 171,000 particles (Table 1 and Extended Data Fig. 2a–f). This yielded a high-quality CMGE core; however, the feature decorating the N-terminal tier was present in only a subset of particles, and, when it was found, it appeared highly dynamic, resulting in poorly defined density (Extended Data Fig. 3a). The N-terminal feature was improved with focused classification on a subset of sCMGE binned particles (Methods and Extended Data Fig. 3b,c). This approach yielded a structure at 4.5-Å average resolution and enhanced quality for the N-terminal MCM-interacting feature. We threaded[20] the yeast sequence onto the available crystal structure of *Xenopus laevis* Mcm10 (ref. [12]), containing a DNA-binding OB-fold and a ZnF domain conserved from yeast to humans (Extended Data Fig. 4a)[21]. This atomic model matched the size and shape of the unassigned density, interacting with the Mcm2, Mcm5 and Mcm6 ZnF domains (Fig. 2d). Docking solutions into this density feature were ranked and compared with available crosslinking mass spectrometry data from a reconstituted CMG–Mcm10 complex[15]. When we mapped crosslinks among Mcm10, N-terminal Mcm2 and Mcm5, we observed that the highest cross-correlation solution best satisfied the distance constraints imposed by the disuccinimidyl suberate crosslinker used in the mass spectrometry experiment (Extended Data Fig. 4b). These results were reproduced when using the AlphaFold2 (ref. [22]) prediction for yeast Mcm10 (Fig. 2d), which provides confidence in our molecular assignment. According to the resulting sCMGE–Mcm10 (hereafter sCMGE10) model, the ZnF domain of Mcm10 interfaces Mcm2 with an arrangement reminiscent of the Mcm4/6–Mcm2 ZnF dimerization across MCM rings in the double hexamer (Fig. 3a), suggesting that Mcm4/6 and Mcm10 use the same mechanism to engage the ZnF binding site on Mcm2. This makes concomitant binding of Mcm10 and Mcm4/6 to Mcm2 impossible. Furthermore, when superposing sCMGE10 onto dCMGE, we noted that Mcm10 would clash with the second CMGE complex in the dCMGE (Extended Data Fig. 4c). In our previous study on dCMGE assembly, we reported a high degree of flexibility between the two CMGE monomers, which could explain how N-terminal Mcm10 finds its way in between the MCM dimerization interface. Conversely, the steric impediment that would prevent N-terminal Mcm10 from engaging MCM is more pronounced in the rigid double hexamer MCM structure (Fig. 3a–c), indicating that N-terminal MCM engagement by Mcm10 must occur after dCMGE formation.

Collectively, our observations invite a model whereby Mcm10 binding to the MCM N-terminal homo-dimerization interface splits the dCMGE into two sCMGE10 complexes. sCMGE prevents re-dimerization of the two MCM motors when DNA translocation is activated. This makes dCMGE splitting irreversible, which favors helicase crossing and origin DNA unwinding.

## Mcm10 ssDNA binding dispensable for replication

We asked whether N-terminal Mcm10 plays a structural role only during initiation, acting as a wedge that splits the dCMGE, or, rather, the ssDNA binding function contained in the OB-fold domain[12] plays a role in the initiation process. To address this question, we first reverted to atomic modeling. We superposed the yeast Mcm10 AlphaFold model to a structure of the *Xenopus* Mcm10 N-terminal domain co-crystallized with ssDNA[12]. This allowed us to identify three residues (F230, Y250 and F247) in the yeast Mcm10 OB-fold that are predicted to interact with ssDNA (Extended Data Fig. 4d). Unlike for the wild-type protein, no ssDNA binding activity could be detected for an Mcm10 variant

containing a triple F230A, Y245A, F247A change (Mcm10 3A; Extended Data Fig. 1a,c). However, dCMGE splitting was as efficient for Mcm10 3A as for the wild-type protein (Extended Data Fig. 4e). Also, DNA synthesis and fork rate levels were similar when using either wild-type or the Mcm10 3A mutant (Extended Data Fig. 4f). This indicates that the ssDNA binding function of Mcm10 is not required for replication initiation or for modulating the rate of replication fork progression.

## The role of C-terminal Mcm10

Our results so far explain how N-terminal Mcm10 binding to N-terminal MCM plays a structural role, making dCMGE separation irreversible. They do not, however, explain how Mcm10 can wedge itself into the relatively tight dimerization interface of dCMGE. We reasoned that a second Mcm10 docking site on the helicase would increase affinity of the interaction and facilitate N-terminal Mcm10 binding in between two N-terminal MCM hexamer sides. Such an element could not be unambiguously resolved in our cryo-EM map, either because of inherent structural flexibility or because two CMGE binding elements in Mcm10 act sequentially and are mutually exclusive. Nonetheless, when we revisited published crosslinking mass spectrometry data on a purified CMG–Mcm10 complex, we noted that the C-terminal and not the N-terminal domain of Mcm10 provides the most extensive interaction with CMG[15]. In fact, a domain including the first 357 residues, which encompass OB and ZnF elements previously discussed, engages in relatively sparse contacts with the MCM ring. The shorter Mcm10 C-terminal domain (residues 358–571) instead accounts for 56% of the MCM crosslinks[15]. To help visualize this interaction, we mapped CMGE crosslinks on a full-length yeast Mcm10 AlphaFold model where unstructured elements were manually stretched for clarity. This interaction map shows that the N-terminal region of Mcm10 engages the N-terminal tier of MCM, and the C-terminal region of Mcm10 contacts the C-terminal tier of MCM (effectively stapling the ATPase domains of Mcm6 and Mcm2 together; Fig. 4a). These observations are compatible with our hypothesis that the Mcm10 C-terminal docking site on the MCM ATPase plays an important function in the initial MCM engagement of the dCMGE, strengthening the interaction with the helicase and facilitating N-terminal Mcm10 access to the Mcm2 ZnF at the dCMGE dimerization interface. To test this model, we generated an Mcm10 mutant lacking residues 358–571 (Mcm10^ΔCTD). This Mcm10 variant retains the ability to bind ssDNA with nanomolar affinity, in the same range observed for similar *Xenopus* Mcm10 constructs used in protein–DNA co-crystallization studies[21,23]. Thus, any change in efficiency of CMGE engagement or splitting should be ascribed to the absence of C-terminal Mcm10 and not to N-terminal Mcm10 misfolding (Extended Data Fig. 1a,c). Next, we assembled constitutively monomeric CMGE at origin DNA using the MCM^Mcm2 6A variant introduced before and supplemented it with either wild-type Mcm10 or Mcm10^ΔCTD. We found that wild-type Mcm10, but not Mcm10^ΔCTD, decorates the N-terminal side of MCM in the sCMGE^Mcm2 6A (Fig. 4b and Extended Data Fig. 1i). These data support the notion that Mcm10 C-terminus drives CMGE engagement of Mcm10. In agreement with this observation, we found that Mcm10^ΔCTD is unable to split dCMGEs reconstituted at origins when used in the low nanomolar concentration range employed for electron microscopy (Fig. 4c,d and Extended Data Fig. 1j). It also does not support DNA replication in vitro in the same concentration range used for the wild-type protein[14] (Fig. 4e). Our results agree with previous observations in cells indicating that C-terminal Mcm10 engagement to CMG supports the splitting of a CMGE dimer at origins[24]. However, if the role of C-terminal Mcm10 is docking onto CMG to increase the Mcm10 affinity, we reasoned, it should be possible to achieve replication initiation by titrating up the Mcm10^ΔCTD concentration (although to such high levels that would make single-particle electron microscopy analysis impossible). Indeed, we found that increasing Mcm10^ΔCTD concentration from the standard 10–20 nM to the 50–200 nM concentration range achieved a short but detectable DNA synthesis product, reflecting replisome

**Table 1 | Cryo-EM data collection, refinement and validation statistics**

| | 'Consensus-sCMGE10' (EMD-17459), (PDB 8P63) | 'ssDNA-sCMGE10' (EMD-17458), (PDB 8P62) | 'nexus-sCMGE10' (EMD-17449), (PDB 8P5E) |
|---|---|---|---|
| **Data collection and processing** | | | |
| Magnification | 130,000× | 130,000× | 130,000× |
| Voltage (kV) | 300 | 300 | 300 |
| Electron exposure (e-/Å$^2$) | 50 | 50 | 50 |
| Defocus range | −1.8 to −3.3 mm | −1.8 to −3.3 mm | −1.8 to −3.3 mm |
| Pixel size (Å) | 1.08 | 1.08 | 1.08 |
| Symmetry imposed | C1 | C1 | C1 |
| Initial particle images (no.) | 20,16,248 | 20,16,248 | 20,16,248 |
| Final particle images (no.) | 1,72,552 | 87,445 | 85,107 |
| Map resolution (Å) | 3.7 | 3.9 | 3.9 |
| FSC threshold | 0.143 | 0.143 | 0.143 |
| **Map resolution range (Å)** | | | |
| **Refinement** | | | |
| Initial model used (PDB code) | 7QHS | 7QHS | 7QHS |
| Model resolution (Å) | 3.5 | 3.5 | 3.5 |
| FSC threshold | 0.5 | 0.5 | 0.5 |
| **Moldel resolution range (Å)** | | | |
| Map sharpening B-factor | −50 | −30 | −30 |
| **Fit to map** | | | |
| Refinement program | Phenix | Phenix | Phenix |
| CC (mask) | 0.79 | 0.78 | 0.8 |
| CC (box) | 0.84 | 0.85 | 0.87 |
| CC (peaks) | 0.72 | 0.71 | 0.74 |
| CC (volume) | 0.77 | 0.77 | 0.8 |
| Mean CC for ligands | 0.75 | 0.79 | 0.78 |
| **Model composition** | | | |
| Non-hydrogen atoms | 52,960 | 53,042 | 53,422 |
| Protein residues | 6,620 | 6,631 | 6,634 |
| Nucleotides | 9 | 9 | 26 |
| Ligands | 16 | 16 | 16 |
| **B-factors, mean (Å$^2$)** | | | |
| Protein | 93.8 | 119.98 | 119.22 |
| Nucleotide | 97.64 | 131.12 | 305.6 |
| Ligand | 80.62 | 107.24 | 109.03 |
| **Root-mean-square deviations** | | | |
| Bond lengths (Å) | 0.002 | 0.002 | 0.003 |
| Bond angles (°) | 0.54 | 0.535 | 0.548 |
| **Validation** | | | |
| Molprobity score | 1.62 | 1.67 | 1.76 |
| Clashscore | 6.76 | 7.04 | 7.86 |
| Rama-Z (whole) | 0.08 | 0.16 | −0.06 |
| Poor rotamers (%) | 0 | 0 | 0 |
| **Ramachandran plot** | | | |
| Favored (%) | 96.36 | 95.86 | 95.24 |
| Allowed (%) | 3.61 | 4.1 | 4.73 |
| Outliers (%) | 0.03 | 0.03 | 0.03 |

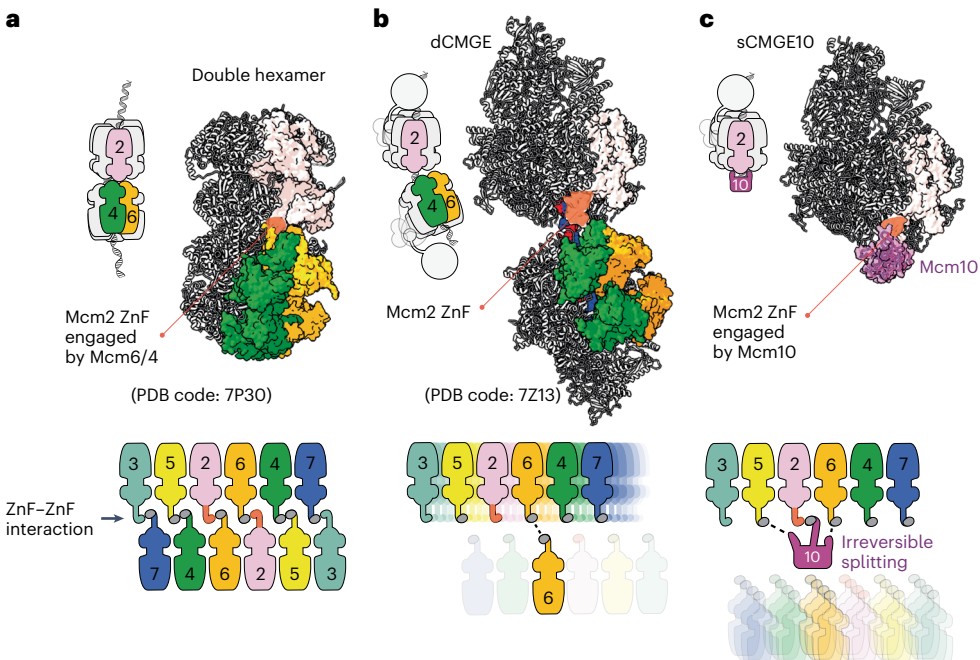

**Fig. 3 | N-terminal Mcm2 exposed as double hexamer transitions to dCMGE.**
**a**, In the DNA-loaded double hexamer, the Mcm2 ZnF domain is buried at the dimerization interface through an interaction with Mcm4 and Mcm6 from the opposed ring. **b**, Double hexamer to dCMGE transition causes a one-subunit register shift, so that the N-terminal Mcm2 becomes exposed. **c**, Mcm10 OB-fold and ZnF domains interact with Mcm6-2-5. Mcm10 ZnF mimics the Mcm2–Mcm6 dimerization interaction between opposed MCM rings.

activation, although inefficient and with low fork rate (Fig. 4f). To establish whether this is indeed the product of a functional replication fork, we asked whether fork rate stimulation could be rescued by adding the Mrc1–Tof1–Csm3 (MTC) complex in *trans*. We found that the DNA synthesis product is extended when MTC is present, indicating that replisomes activated with Mcm10$^{\Delta CTD}$ are responsive to fork rate stimulation factors (Fig. 4f). In summary, truncating the Mcm10 C-terminal domain decreases the affinity for CMGE and abrogates the fork rate stimulation function but does not block origin activation.

## Separable Mcm10 functions in initiation and elongation

We showed that truncating the C-terminal domain lowers the affinity of Mcm10 for CMGE, decreasing dCMGE splitting efficiency and replication. When DNA synthesis products are detected, the length is short, indicating loss of the fork rate stimulation function. A similar effect on fork rate was reported when Mcm10 is phosphorylated by kinase Rad53 in vitro, recapitulating a key reaction in the DNA damage response pathway. Unlike C-terminal domain truncation, however, Mcm10 phosphorylation by Rad53 does not affect initiation efficiency[17]. These observations suggest that dCMGE splitting and N-terminal sCMGE decoration by Mcm10, which we interpret as initiation-specific events, should not be affected by Rad53 phosphorylation. To test our hypothesis, we first ensured that pre-phosphorylation of Mcm10 by Rad53 resulted in a detectable shift in electrophoretic mobility, indicative of robust phosphorylation by the wild-type protein but not the catalytically dead variant (Extended Data Fig. 5a,b). Using these proteins, we confirmed previous observations that Mcm10 phosphorylation, but not incubation with catalytically dead Rad53, severely affects DNA replication in time-course experiments[17] (Extended Data Fig. 5c,d). This defect was robustly rescued by addition of Mrc1, 20 min after replication initiation (Extended Data Fig. 5e), implying that phosphorylation affects the rate of replication fork progression and not the efficiency of replication initiation[17]. Using the same preparations, we assembled dCMGE for negative-stain electron microscopy analysis and then added Mcm10 to probe its ability to split the complex into

two sCMGEs (Extended Data Fig. 5f). We found that neither Rad53 phosphorylation nor co-incubation with catalytically dead Rad53 had any effect on the ability of Mcm10 to split the dCMGE. N-terminal MCM decoration by Mcm10 in sCMGE was equally unaffected (Extended Data Fig. 5f–h). Collectively, our data indicate that dCMGE splitting and N-terminal sCMGE decoration are replication initiation functions of Mcm10, which remain unchanged upon phosphorylation by Rad53. Mcm10 C-terminal domain truncation and Mcm10 phosphorylation by Rad53 have a similar inhibitory effect on fork rate stimulation, which can be rescued by the MTC complex in both conditions. These observations indicate that the fork rate stimulation function resides in the Mcm10 C-terminal domain. Future efforts should establish whether phosphorylation sites that block fork rate stimulation map within the Mcm10 C-terminus. Conversely, phosphorylation by Rad53 does not appear to affect the global affinity of the Mcm10–CMGE interaction, given that dCMGE splitting, N-terminal MCM decoration and initiation efficiency remain unperturbed upon Mcm10 phosphorylation. This is consistent with in vivo evidence that Rad53 blocks initiation by phosphorylating Dbf4 and Sld3, not Mcm10 (ref. 25).

## Duplex DNA unwinding from N-terminal MCM

Our results so far establish two separable roles for the C-terminal Mcm10 domain: (1) stimulating fork rate, which can be modulated by the Rad53 kinase, and (2) docking onto the helicase, which supports splitting of the dCMGE through N-terminal Mcm10 access to N-terminal MCM. The latter is insensitive to phosphorylation by Rad53. The N-terminal Mcm10 interaction with N-terminal MCM is poised to drive dCMGE splitting and prevent MCM re-dimerization when the two helicases translocate toward one another, instead favoring helicase crossing. A second requirement for helicase crossing is that the two DNA strands trapped within each MCM ring are unwound, and the lagging strand is ejected, so that the strand expelled from one helicase becomes the translocation strand of the other helicase. Upon dCMGE formation, only a small DNA bubble is nucleated within the core of each MCM ATPase, and 1.5 turns of double helix per helicase remain to be

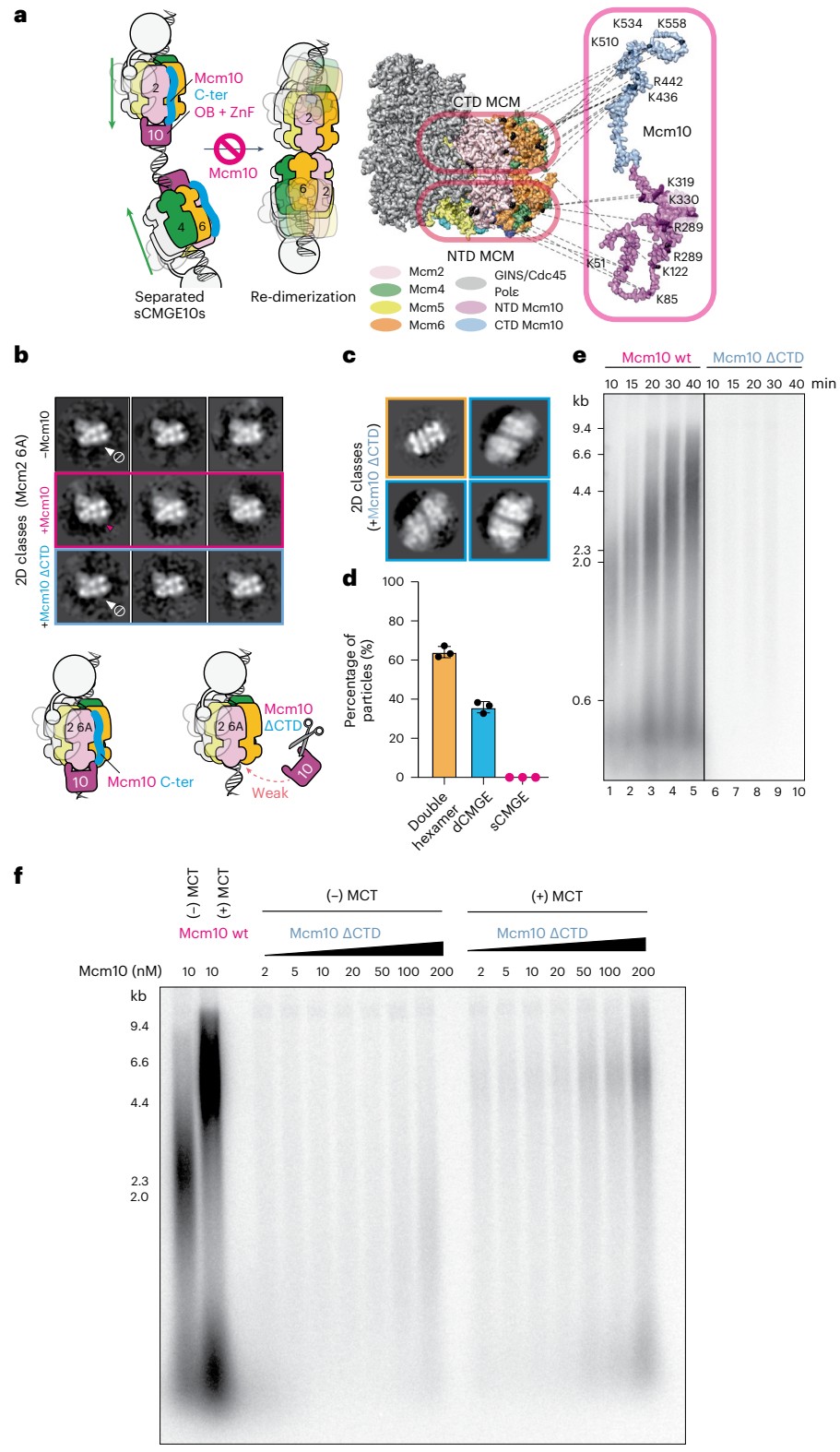

**Fig. 4 | C-terminal Mcm10 supports, but is not essential for, origin activation.**
**a**, Available crosslinking-coupled mass spectrometry data mapped on the sCMGE–Mcm10 structure. The N-terminus of Mcm10 binds to the N-terminal tier of MCM, whereas the C-terminus of Mcm10 binds to the C-terminal tier of MCM. **b**, C-terminal Mcm10 truncation fails to decorate N-terminal MCM in an Mcm10-independent sCMGE variant at concentrations compatible with electron microscopy analysis. **c**, C-terminal Mcm10 truncation fails to split the dCMGE at concentrations compatible with EM analysis. **d**, Quantitation of particles contributing to 2D averages shown in **c**. The experiment was performed three times. Error bars, mean ± s.d. **e**, C-terminal Mcm10 truncation does not

support DNA replication reconstituted in vitro when tested at low nanomolar concentration. The experiment was performed three times. **f**, C-terminal Mcm10 truncation supports replication initiation when probed within the 50–200 nM concentration range. However, the fork stimulation function is absent. Rate stimulation can be recovered by supplementing the MTC complex in *trans*. This demonstrates that, despite the low affinity and the loss of fork rate stimulation, the isolated N-terminal domain is the only element required to initiate DNA replication. The fork stimulation function, instead, is not essential for initiation. The experiment was performed twice. wt, wild-type.

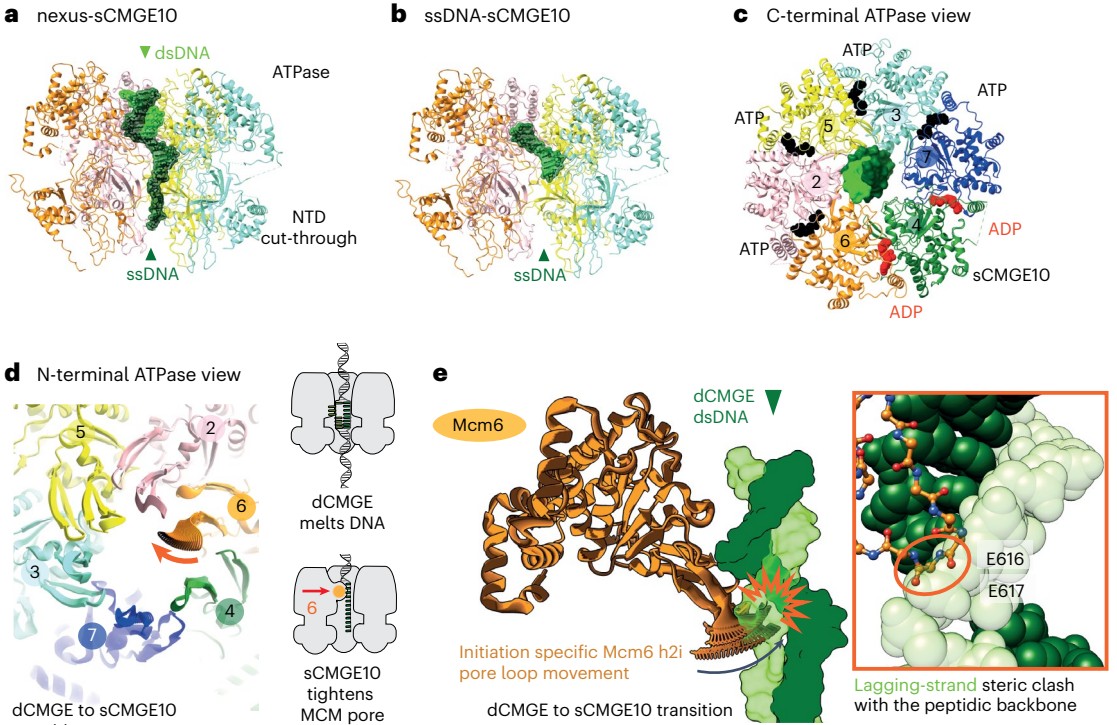

**Fig. 5 | DNA engagement by CMGE10. a**, sCMGE10 class containing duplex DNA bound by the ATPase domain and ssDNA running through the N-terminal MCM tier. **b**, sCMGE10 particle set containing ssDNA contacted by MCM ATPase pore loops. **c**, Nucleotide occupancy within the ATPase tier viewed from the C-terminal MCM side. **d**, ATPase pore loops viewed from the N-terminal MCM side. Interpolated structures derived from a molecular morph highlight pore loop changes in the transition between dCMGE and sCMGE10. Reconfiguration of Mcm6 h2i pore loop is specific to the origin unwinding structures reported in this work. **e**, CMGE10-specific Mcm6 h2i pore loop positioning constricts the MCM channel, making duplex DNA engagement within the N-terminal MCM channel impossible. NTD, N-terminal domain.

unwound. Whether Mcm10 expands the DNA bubble by promoting DNA unwinding toward the N-terminal or the C-terminal side of MCM first is unknown. To investigate this, we focused our analysis on the interactions between helicase and DNA in our cryo-EM reconstruction. In the consensus structure, sCMGE engages ssDNA. During early three-dimensional (3D) classification efforts, however, we detected density suggestive of a short duplex DNA segment found in the MCM C-terminal pore. To improve the duplex DNA density, we exploited a strategy previously developed to enrich for DNA-interacting MCM. This protocol employs subtraction of the signal corresponding to the C-terminal half of MCM, followed by two-dimensional (2D) classification without alignment to peek inside the ATPase tier[8]. We focused our analysis on sCMGE10 side views that clearly show DNA emerging from the residual N-terminal domain of MCM. With this strategy, we could separate single-stranded and double-stranded DNA-interacting particles based on 2D averages. We then combined non-subtracted, duplex-DNA-enriched side views with all other views and subjected this particle subset to 3D classification. Enriching for duplex-engaged side views achieved successful separation of ssDNA-engaged sCMGE10 from sCMGE10 encircling a duplex ssDNA nexus, with both structures refined to an average resolution of 3.9 Å (Table 1 and Extended Data Figs. 2g–l and 3b). In the latter structure, hereafter referred to as nexus-sCMGE10, duplex DNA is entrapped in the ATPase tier, and ssDNA projects toward the N-terminal tier of MCM. We interpret the nexus-sCMGE10 as a helicase that has unwound only one of the 1.5 turns of double helix found in each MCM hexamer of the dCMGE. (Fig. 5a and Extended Data Video 1). We cannot determine whether the unwound lagging strand is invisible because it is flexible or because strand ejection has started at this stage. We interpret ssDNA-sCMGE10, instead, as a helicase that has fully unwound and ejected the lagging strand

(Fig. 5b and Extended Data Video 2). Our interpretation is supported by previous western blot analysis of the same biochemical conditions imaged in our cryo-EM experiment, which revealed origin recruitment of single-stranded binding factor replication protein A (RPA)[10]. This can possibly occur only when ssDNA is found outside of the MCM helicase ring. Together, our structures of two intermediates of the origin activation reaction indicate that DNA unwinding is initiated from the N-terminal side of MCM in the sCMGE10 complex.

**Pore loop movement discriminates translocation strand**

So far, we have explained how Mcm10 binding to MCM splits the dCMGE and promotes unwinding of DNA inside the MCM central channel. To achieve origin DNA unwinding, each sCMGE10 particle must transition from encircling two DNA strands to encircling only one strand. But how can MCM discriminate between the strand to be retained inside the hexamer pore and the strand to be ejected? To address this issue, we compared the dCMGE reconstruction containing partially melted duplex DNA with the structure of nexus-sCMGE10, containing one turn of double helix unwound (or of ssDNA-sCMGE10, where DNA fully unwound). Although it is established that ATP hydrolysis is needed for DNA unwinding and strand ejection[10], we noted that all structures display the same nucleotide occupancy in the ATPase sites (Extended Data Fig. 6a). This indicates that, after unwinding one or the full 1.5 turns of duplex DNA, the sCMGE10 ATPase resets to the same lower-energy state observed in dCMGE (Fig. 5c). Although nucleotide occupancy within ATP hydrolysis sites is the same as in the dCMGE, we noticed one change in the structure of the ATPase domain occurring upon transition from dCMGE to the nexus-sCMGE10 structure (or to the ssDNA-sCMGE10 structure). Here, the Mcm6 helix 2 insert (h2i) pore loop, which projects from the ATPase domain, moves from

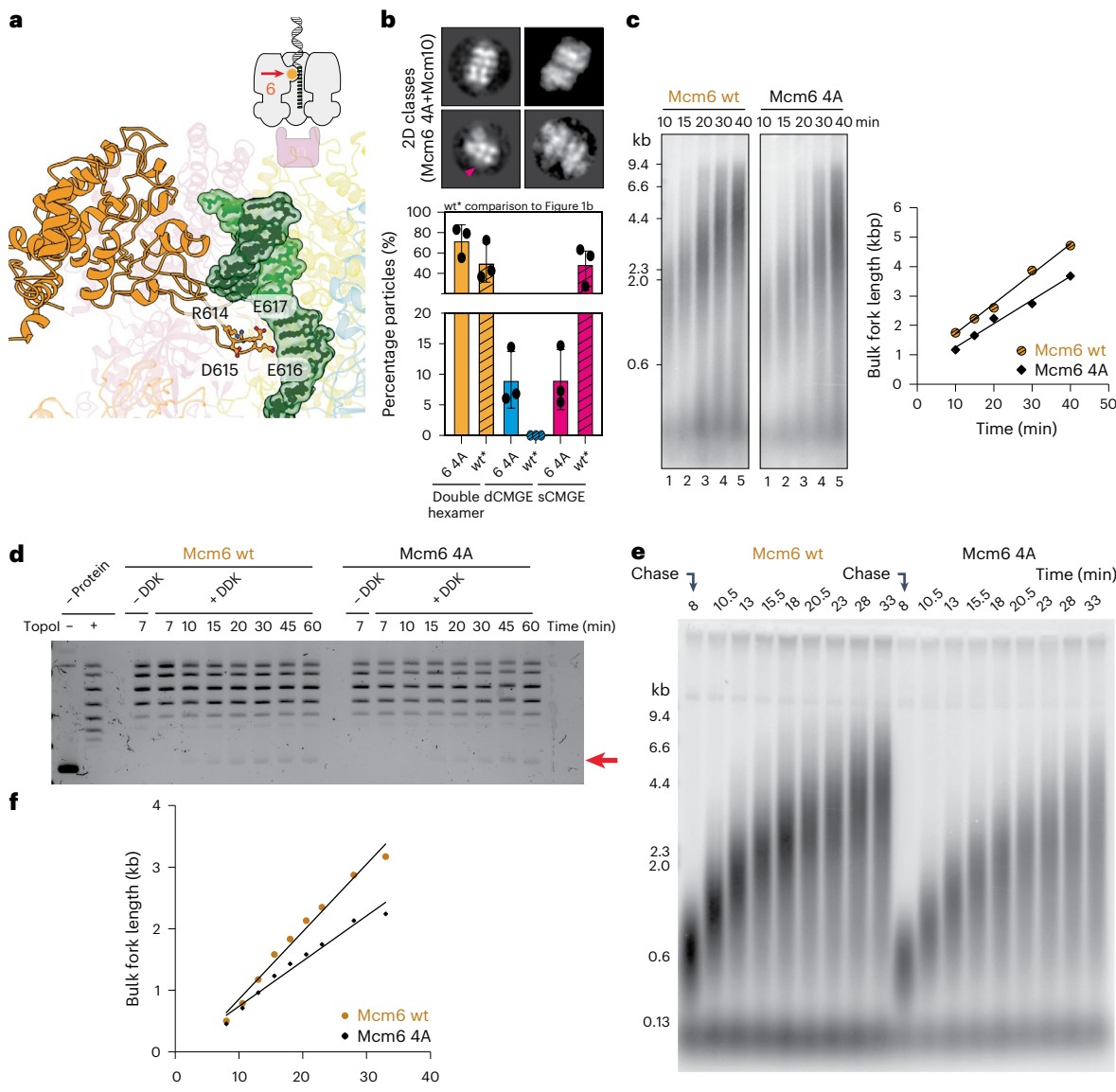

**Fig. 6 | Mcm6 makes new leading-strand contacts. a**, CMGE10-specific contacts between Mcm6 h2i and the leading-strand template include R614, D615, E616 and E617. **b**, Mutation of these four residues (Mcm6 4A) negatively affects dCMGE splitting by Mcm10, revealing an origin activation defect. In fact, only 50% of Mcm6 4A CMGE-containing particles are sCMGEs, which corresponds to a double-to-single CMGE conversion efficiency of 33%. The experiment was performed three times. Error bars, mean ± s.d. Shading marks comparison with Fig. 1b. **c**, An MCM-Cdt1 variant containing the Mcm6 4A changes can be loaded to wild-type levels (Extended Data Fig. 7a) but results in impaired DNA replication as observed in a time-course experiment. The experiment was performed twice. **d**, The same defective kinetics is observed for the Mcm6 4A in the time-course experiment of a plasmid-based DNA unwinding assay. The experiment was performed twice. **e**, A pulse-chase assay reveals an elongation defect for the Mcm6 4A mutant, in addition to the initiation defect detected with the dCMGE splitting assay shown in **b**. **f**, Quantitation is the quantitation of **e**. The experiment was performed twice. wt, wild-type.

the side to the center of the helicase channel (Fig. 5d and Extended Data Video 3). An overlay with the dCMGE structure reveals that the Mcm10-dependent pore loop movement constricts the MCM ring central channel, which becomes too narrow to harbor duplex DNA. In particular, the Mcm6 h2i movement creates a steric clash between the peptide backbone of residues E616–E617 and the lagging-strand (but not the leading-strand) template (Fig. 5e and Extended Data Video 3). Narrowing of the ATPase channel of CMG can be observed only in our activation reactions using the reconstituted cellular pathway on a native origin DNA sequence. It was not observed in previous structures where pre-formed CMG (assembled through overexpression in cells) was threaded through artificial substrates, such as a model DNA fork that mimics CMG during elongation[26–30]. Thus, pore narrowing appears specific to initiation of origin DNA unwinding.

Not only the initiation-dependent movement of Mcm6 h2i is incompatible to lagging-strand retention inside the MCM pore, close inspection of the nexus-sCMGE10 and ssDNA-sCMGE10 structures reveals that this movement also leads to the establishment of new contacts (not seen in dCMGE[31]) with the leading-strand template that becomes the translocation strand upon replication fork establishment. To test whether these new interactions play any role during initiation and fork progression, we generated an MCM variant (Mcm6 4A) targeting four Mcm6 h2i residues (R614A, D615A, E616A and E617A) that bind (or map very close to) ssDNA downstream of the duplex ssDNA nexus in the unwinding intermediate, nexus-CMGE10 (Fig. 6a and Extended Data Fig. 6b–g). To identify any initiation defect, we performed negative-stain electron microscopy analysis. We found that double hexamers can still be loaded with MCM[Mcm6 4A], and Mcm10

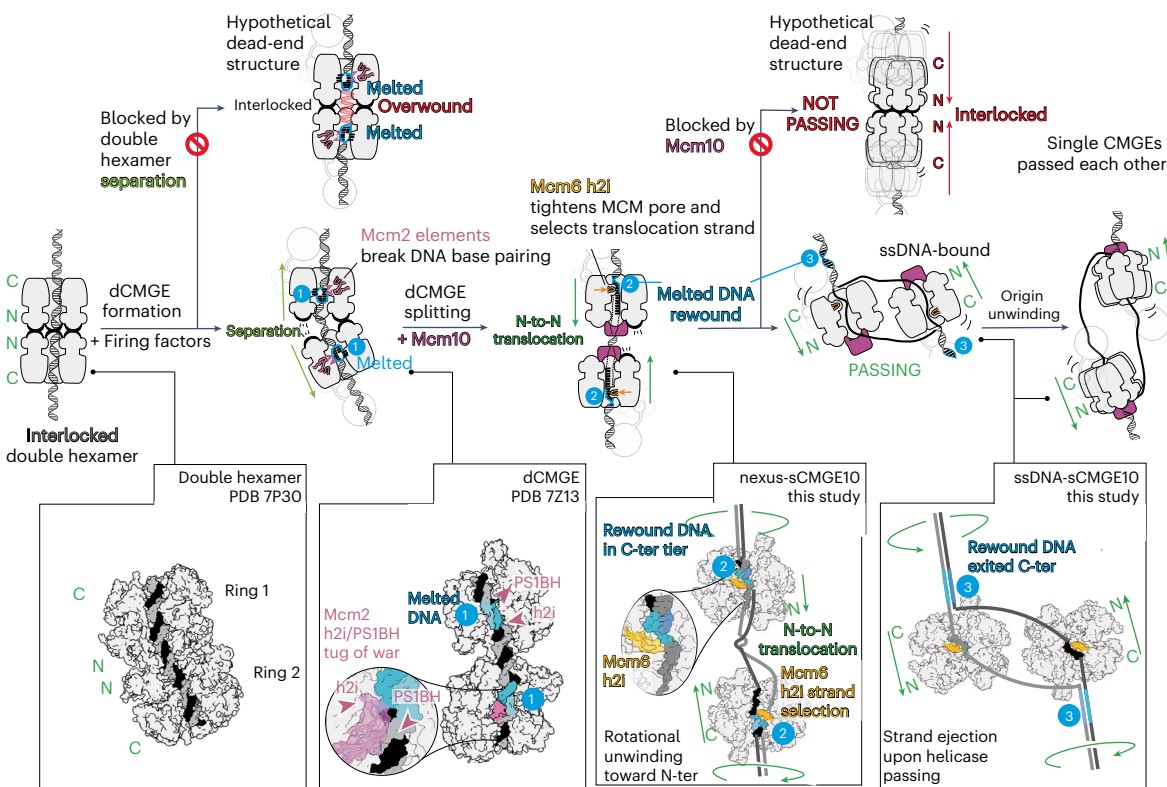

**Fig. 7 | Cartoon representation of the steps that lead to replication origin firing.** Two MCMs are loaded onto duplex DNA as a double hexamer with interlocked dimerizing N-termini (left). dCMGE formation causes Mcm2 ATPase pore loops to engage DNA and break base pairing, nucleating DNA melting within each ATPase tier (mid-left). To prevent DNA overwinding, the two helicases partially separate while exposing duplex DNA in between (mid-right). Mcm10 recruitment splits the dCMGE via an N-terminal interaction with N-terminal MCM. It also allosterically alters the helicase structure. The Mcm6 h2i pore loop moves toward the MCM central channel. In this new position, Mcm6 h2i would

clash with the lagging-strand template encircled by dCMGE. This movement disrupts Mcm2 pore loop contacts and establishes new interactions with the translocation strand. Mcm10-dependent ATPase activation triggers DNA unwinding in the MCM N-terminus. In this configuration, the two helicases can translocate toward another (right). Helicase passing favors lagging-strand ejection and full origin unwinding. DNA melted in dCMGE (position 1) is rewound and trapped within the ATPase in nexus-sCMGE10 (position 2) and fully rewinds behind the helicase as the helicases cross paths (position 3). DNA in this configuration is ready to become unwound by the opposed helicase incoming.

still decorates sCMGEs in reconstituted origin unwinding reactions; however, the efficiency of dCMGE splitting drops by 66% (Fig. 6b and Extended Data Fig. 7a–c), indicating inefficient origin activation. We also performed a time-course experiment to find that DNA replication with the Mcm6 4A variant is partially impaired (Fig. 6c; repeat shown in Extended Data Fig. 7d). The same kinetic defect was observed in a plasmid-based time-course assay that monitors DNA unwinding (Fig. 6d). A second unwinding experiment where ORC concentration was increased from 10 nM to 30 nM shows the same trend (Extended Data Fig. 7e). To establish whether this defect reflects only the inefficient origin activation detected by electron microscopy or also an elongation defect, we performed a pulse-chase experiment to compare the Mcm6 4A variant with the wild-type protein. During pulse-chase analysis, the extension of DNA synthesis products labeled in the first few minutes is monitored independently of the kinetics of initiation. The Mcm6 4A fork rate is lower than the wild-type protein, indicating a clear elongation defect (Fig. 6e,f). Although evident, the relatively small effect that the Mcm6 4A mutant has can be rationalized, given that 34 additional contacts help select the leading-strand template in the helicase ring of the nexus-sCMGE10 structure (Extended Data Fig. 6b). In summary, our results explain how structural changes upon dCMGE-to-sCMGE10 transition lead to ejecting the lagging strand, given the clashes with the peptide backbone of Mcm6 h2i, and selecting the leading strand for translocation via a set of new side chain contacts that support both initiation and elongation.

## Discussion

Studies in yeast established that two MCM helicases are loaded onto origin DNA as a catalytically inactive double hexamer with dimerizing N-termini[5–8,32] and a universally conserved inter-hexamer register[33]. Helicase activation later occurs in two steps: DNA melting nucleated upon dCMGE formation, followed by origin DNA unwinding triggered by Mcm10 and supported by ATP hydrolysis[10]. A cryo-EM study of the dCMGE formation step revealed that two ATPase pore loops within the same Mcm2 subunit engage in a molecular tug of war that breaks DNA base pairing but prevents further unwinding (Fig. 7)[11]. Upon double hexamer to dCMGE transition, the Mcm2 Pre-Sensor 1 pore loop establishes a new DNA grip that untwists the leading-strand template away from the double hexamer dimerization interface. During the same transition, the Mcm2 h2i pore loop maintains one protein–DNA interaction (already established in the double hexamer), pushing the lagging-strand template toward the dimerization interface. To compensate for this structural change and prevent DNA overwinding between the two helicases, the two MCM rings move away from one another, remaining tethered via Mcm6 while they expose a segment of duplex DNA in the space created with double hexamer separation[11]. To select the translocation strand and establish unidirectional DNA unwinding with 3′-to-5′ polarity from N-terminal to C-terminal MCM[10,34,35], a change must occur inside the ATPase ring, which causes the helicase motor to let go of the lagging strand and grip the leading strand alone. This, we found, is what happens when Mcm10 recruitment

allosterically alters the MCM ring structure. Here, the h2i pore loop of Mcm6 moves toward the center of the MCM channel, occupying a position that would clash with the lagging-strand template. This change dislodges the neighboring Mcm2 h2i DNA interaction, releasing the lagging strand from the MCM grip (Fig. 5e). At the same time, Mcm6 h2i establishes new contacts with the leading-strand template, which contribute to selecting it as the translocation strand (Fig. 6a). As Mcm10 triggers ATP-hydrolysis-powered translocation, the leading strand must rotate around the lagging strand to unwind one turn of DNA within N-terminal MCM (Fig. 7). Our nexus-sCMGE10 structure supports this model, as it shows unwound DNA at the N-terminus and a stretch of duplex DNA still entrapped by the C-terminal ATPase tier (Fig. 5a). DNA translocation via hand-over-hand sequential rotary cycling (the accepted mechanism for hexameric ATP-powered translocases[26,36–39]) involves threading of the leading-strand template from the N-terminal to the C-terminal domain of MCM during unwinding. The persistence of a stretch of duplex DNA entrapped within the ATPase in our nexus-sCMGE10 structure predicts that the N-terminally unwound DNA will rewind as it enters the ATPase tier and the helicase moves forward (Extended Data Video 3). Our observations suggest that, no matter how far it advances, one lone sCMGE10 helicase might never fully eject the lagging-strand template from the MCM ring pore. Instead, two motors moving past one another would be needed to strip the lagging strand out of the opposed DNA ring pore (Fig. 7). Although we cannot exclude that strand ejection occurs before helicases cross paths via a transient event that we still do not understand, we note that our conclusion agrees with previous observations by Stephen P. Bell's laboratory. In the Bell laboratory study[40], an Mcm10-activated CMGE variant (assembled from a single, DNA-loaded MCM hexamer that cannot form double hexamers) starts untwisting DNA but does not support full DNA unwinding through lagging-strand ejection. Mechanistic models proposed for different replicative helicase systems (the archaeal MCM[37], the SV40 LtAg[41] and the metazoan-specific double CMG-DONSON initiation complex[42]) also invoke two motors moving toward one another to achieve full DNA unwinding. These models envisage that each of two juxtaposed motors translocating with 3′ to 5′, N-terminal to C-terminal polarity would pull DNA away from the opposed hexameric ring, until one of the two strands becomes ejected from each helicase pore[37,41]. Work by the O'Donnell laboratory[43], published while this paper was under revision, provides clear evidence in support of our model, by showing that that two single converging yeast CMGs melt and fully unwind the double helix via the DNA shearing mechanism described above. The authors found that Mcm10 stimulates, but is not required for, duplex DNA opening, which does not explain why Mcm10 is essential for replication initiation, as instead established in our earlier study on origin activation reconstituted in a test tube with yeast proteins[10]. The cryo-EM work presented here fills this gap in knowledge, as it monitors in vitro reconstitution of replicative helicase activation from its initial loading as an MCM double hexamer, to nucleation of DNA melting within the dCMGE, all the way through to origin unwinding[10,44]. This is how we found that the N-terminal domain of Mcm10 plays an essential structural role promoting the splitting of the double CMGE by directly engaging the N-terminal homo-dimerization interface of MCM, which becomes partially exposed as the two MCM hexamers start to separate. We found that the ssDNA binding function of Mcm10 is not essential for initiation, possibly because this activity is redundant with RPA. Our finding agrees with the striking observation that Mcm2 variants containing amino acid changes affecting the N-terminal ZnF domain bypass the requirement for Mcm10 (ref. 13). As we established here that Mcm10 plays an essential structural role in splitting the dCMGE, it could be that the Mcm10 bypass mutants contribute to weakening the dCMGE interface and promote splitting into two sCMGEs. Compatible with previous results[45], we also found that the fork rate stimulation function is dispensable for origin activation and that it can be suppressed by Rad53

with no effect on dCMGE splitting. Other structural aspects observed in our study, such as the narrowing of the MCM channel observed in our sCMGE10 structures, might be specific to the origin activation reaction or could still transiently occur during duplex DNA shearing by two converged CMG helicases. Although our work provides critical insights into DNA replication initiation, several aspects of Mcm10 function remain to be investigated. For example, do two Mcm10-activated sCMGEs rotate with respect to one another[11], and do they interact as they cross paths? What interface in the MCM ring hexamer opens to facilitate lagging-strand ejection? What is the mechanism for fork rate stimulation by Mcm10? Addressing these questions will be critical to understand replication initiation and elongation.

## Online content

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

## Methods

### Protein purification

ORC, Cdc6, Mcm2–7/Cdt1, DDK, CDK, Sld2, Sld3–Sld7, Cdc45, Dpb11, Polε, Mrc1, Csm3/Tof1, Mcm10 and MH were purified based on established protocols[9–11,44,46–48].

### Cloning, expression and purification of Mcm2–7/Cdt1 mutants

The pMG73 plasmid was generated using a QuikChange Site-Directed Mutagenesis Kit (Agilent) with oMG40 and oMG41 primers (Supplementary Table 1) according to the manufacturer's protocol. The pMG73 (Supplementary Table 2) was integrated into the yAM20 strain (Supplementary Table 3), yielding the yMG44 strain (Supplementary Table 3) that was used to overexpress the Mcm6 4A (Mcm6 R614A D615A E616A E617A) mutant. Mcm6 4A was purified as Mcm2–7/Cdt1 wild-type.

### Cloning, expression and purification of Mcm10 mutants

Mcm10 ΔCTD (Δ358–571) expression vector was cloned into pET302-NT-His (vector sequence provided in the Supplementary Information) using Azenta. T7 express cells (New England Biolabs) were transformed with Mcm10 ΔCTD expression plasmid (pSSH006). Transformant colonies were inoculated into a 250-ml LB culture containing ampicillin (100 µg ml$^{-1}$), which was grown overnight at 37 °C with shaking at 200 r.p.m. The next morning, 2 × 2 L of LB containing ampicillin (100 µg ml$^{-1}$) was inoculated with 100-ml overnight culture. The cultures were grown at 37 °C to optical density at 600 nm (OD$_{600}$) of 0.5. The cultures were then moved to 16 °C, at 200 r.p.m. After 40 min, OD$_{600}$ of 0.6 was reached. To then induce expression, 0.5 mM isopropyl β-D-1-thiogalactopyranoside (IPTG) was added, and cells were left shaking for 3 h. Cells were collected by centrifugation at 4,000 r.p.m. for 20 min. Before lysis, cell pellets were resuspended in 50 ml of lysis buffer (25 mM HEPES pH 7.6, 10% glycerol, 0.02% NP-40, 1 mM EGTA, 500 mM NaCl, Roche protease inhibitor tablets, 1 mM dithiothreitol (DTT) + 0.5 mM Pefabloc). Cells were lysed by sonication for 120 s (5 s on, 5 s off) at 40%. After centrifugation at 20,000 r.p.m. for 30 min, the supernatant was incubated with 1 ml of M2 Flag resin (Sigma-Aldrich). The resin was washed extensively with wash buffer (25 mM HEPES pH 7.6, 10% glycerol, 0.02% NP-40, 300 mM NaCl). Mcm10 ΔCTD was eluted by the addition of 0.25 mg ml$^{-1}$ 3× Flag peptide. Fractions containing Mcm10 ΔCTD were then incubated with 2 ml of Ni-NTA resin (Qiagen). After washing (25 mM HEPES pH 7.6, 10% glycerol, 0.02% NP-40, 300 mM NaCl, 10 mM imidazole), the Mcm10 ΔCTD was eluted using 200 mM imidazole. Protein fractions were concentrated and loaded onto a Superose 6 Increase (24 ml) equilibrated in 25 mM HEPES pH 7.6, 10% glycerol, 0.02% NP-40, 0.5 mM EDTA, 200 mM NaCl, 2 mM DTT. Mcm10 ΔCTD fractions were then pooled, concentrated, aliquoted and snap frozen in liquid N$_2$ (Mcm10 ΔCTD yield, 100 µg).

### Microscale thermophoresis

In a microplate (384-well, F-bottom, Greiner Bio-One), 15-µl reactions were set up, covering a concentration range of 2,122, 1,273, 636, 212, 64, 21, 6.4, 2.1, 0.6, 0.2 and 0 nM Mcm10 wild-type or Mcm10 CTD mutant in reaction buffer (25 mM HEPES-KOH pH 7.6, 100 mM K-glutamate, 10 mM magnesium acetate, 0.02% NP-40, 0.5 mM TCEP). Each 15-µl reaction contained 5 nM fluorescently labeled ssDNA, CCCCCCCCCCCC[FAM]. Reactions were incubated for 30 min before measurements. Microscale thermophoresis measurements were carried out with Monolith NT.115 Premium Capillaries and Monolith NT.115. Using a blue LED excitation, 100% laser power and 20% MST, temporal fluorescence intensity traces were recorded for the different protein concentrations at room temperature using NTControl version 2.2.1. Traces were analyzed using NT Analysis 1.5.41, showing DFluorescence [temperature jump] ≥ 10. Independent biological triplicates were normalized between 0 and 1. Dose–response curves including standard error were plotted in GraphPad Prism version 9.4.1. The binding model used to fit the DNA binding data was as follows: nonlinear regression (curve fit) using a

'specific binding with hill slope' model $Y = Bmax \times X^h / (Kd^h + X^h)$. The data were normalized (from 0 to 1) for comparison purposes, by setting the lowest value in each replicate to 0 and the highest value to 1 (and then plotting the mean value with standard errors).

### DNA template: short 168-bp MH-flanked origins

The native ARS1 origin of replication flanked by M.HpaII was amplified by polymerase chain reaction (PCR) and purified as previously described[9]. MH-flanked origins were prepared based on previously established protocols[9,11].

### In vitro CMG assembly and activation on short MH-flanked origins

CMG assembly and activation were carried out by adapting previously published protocols[11]. In brief, 20 nM ARS1 MH-flanked origin DNA was incubated with 52 nM ORC, 52 nM Cdc6 and 110 nM Mcm2–7/Cdt1 for 25 min at 24 °C in loading buffer (25 mM HEPES-KOH pH 7.6, 100 mM K-glutamate, 10 mM magnesium acetate, 0.02% NP-40, 0.5 mM TCEP) + 5 mM ATP. Next, 80 nM DDK was added to the reaction and incubated for a further 10 min at 24 °C. DNA-bound protein complexes were isolated by incubation with 4 µl of MagStrep 'type3' XT beads (IBA), pre-washed in 1× loading buffer, for 30 min at 24 °C, to pull on twin-strep-tagged MH. Non-DNA bound proteins were removed by washing the beads three times with 100 µl of wash buffer (25 mM HEPES-KOH pH 7.6, 5 mM magnesium acetate, 0.02% NP-40, 500 mM NaCl), followed by one wash with 100 µl of loading buffer. DNA-loaded, phosphorylated double hexamers were eluted in 20 µl of elution buffer (25 mM HEPES-KOH pH 7.6, 105 mM K-glutamate, 10 mM magnesium acetate, 0.02% NP-40, 0.5 mM TCEP, 27 mM biotin, 5 mM ATP) for 10 min at 24 °C. The supernatant was then removed, and 125 nM CDK was added and incubated for 2 min at 30 °C. A mix of firing factors was then added to a final concentration of 45 nM Dpb11, 150 nM GINS (either His-GINS or TwinStrepII-GINS), 120 nM Cdc45, 30 nM Polε, 45 nM Sld3–Sld7 and 75 nM Sld2, including or excluding 22 nM Mcm10 or Mcm10 mutants. After a 14-min incubation, the reaction was applied directly to grids.

### DNA replication assays

Replication assays were carried out as described previously[11] using pJY22 plasmid (Supplementary Table 2). Staged replication reactions containing Rad53, Mrc1 and controls were performed as previously described with the exception of Rad53:Mcm10 1:10 ratio[17]. Mrc1 and Csm3/Tof1 were used at 20 nM concentration. For pulse-chase experiments, the conditions were the same as for standard DNA replication assay, except that the concentration of dCTP in pulse was reduced to 4 µM, whereas, during the chase, it was increased to 600 µM. The chase was at either 8 min (Fig. 6e,f) or 9 min (Extended Data Fig. 4f). Quantitations were performed using ImageJ2/Fiji version 2.3.0.

### Plasmid-based DNA unwinding assay

DNA unwinding assays were carried out using previously published protocols[11]. In brief, the DNA unwinding assay was performed using 3.2-kb pBS_ARS1_WTA plasmid[49] following a published protocol[10]. Then, 25 fmol of plasmid DNA was treated with 20 nM Topo I for 30 min at 30 °C in a buffer containing 25 mM HEPES-KOH pH 7.6, 100 mM K-glutamate, 10 mM magnesium acetate, 0.02% NP-40-S, 5% glycerol, 2 mM DTT, 5 mM ATP. Next, 10 nM ORC, 50 nM Cdc6 and 100 nM Mcm2–7/Cdt1 were added for 20 min at 30 °C. The reaction was then supplemented with 50 nM DDK, and incubation was continued for 30 min at 30 °C. Additional buffer was supplemented to achieve a final concentration of 250 mM K-glutamate, 25 mM HEPES, 10 mM Mg-acetate, 0.02% NP-40-S, 8% glycerol, 400 µg ml$^{-1}$ BSA, 5 mM ATP, 1 mM DTT. 25 nM Topo I. The mix of firing factors was prepared before use and added at time 0, reaching a final concentration of 30 nM Dpb11, 20 nM GINS, 50 nM Cdc45, 20 nM Polε, 20 nM CDK, 10 nM Mcm10, 25 nM Sld3–Sld7, 50 nM Sld2, 50 nM RPA. After a 40-min incubation

at 25 °C, the reaction was quenched using 13 mM EDTA, 0.3% SDS and 0.1 mg ml⁻¹ Proteinase K (Merck) and incubated at 42 °C for 20 min. The sample was extracted once with phenol:chloroform:isoamylalcohol (25:24:1) and ethanol precipitated, and the DNA pellet was resuspended in 1× Tris-EDTA for analysis. Samples were run in 1.5% agarose gel with TAE, followed by staining with ethidium bromide.

### Negative-stain EM sample preparation and data collection

Negative-stain sample preparation was conducted using previously published protocols[11]. Preparation of negative-stain samples was performed on either 300-mesh or 400-mesh copper grids with carbon film (EM Resolution or Agar Scientific, respectively). Grids were glow discharged for 60 s at 25 mA (GloQube Plus, Quorum), and 4 µl of sample was applied to the grids, followed by 2-min incubation. Grids were stained by two successive applications of 4 µl of 2% (w/v) uranyl acetate with quick blotting between the applications. The second stain application was blotted after 20 s to remove excess stain, and grids were stored before imaging. Data collection was carried out on a Tecnai LaB6 G2 Spirit transmission electron microscope (FEI) operating at 120 keV. Two cameras were used for micrograph collection: a 2,000 × 2,000 Gatan Ultrascan 100 camera at a nominal magnification of 30,000 (with a physical pixel size of 3.45 Å per pixel) and a 4,000 × 4,000 Gatan RIO at a nominal magnification of 29,000 (with a physical pixel size of 3.1 Å per pixel). Collections were carried out within a −0.5 µm to −2.0 µm defocus range. Digital Micrograph software was used for data acquisition.

### Negative-stain electron microscopy analysis image processing

Negative-stain electron microscopy analysis image processing was carried out using approaches described previously[11]. A particle subset was manually picked using RELION 3.1 (ref. 50) and used to train a Topaz model for particle picking[51]. Negative stain image processing was performed using RELION 3.1. The contrast transfer function (CTF) was estimated using Gctf[52], and particles were extracted and subjected to reference-free 2D classification in RELION 3.1. The same particle population trends were observed when a different team member re-analyzed the particle stacks performing 2D classification with cryoSPARC[53].

When only sCMGE and double hexamer 2D classes were found, conversion efficiency for sCMGEs was calculated by dividing the number of sCMGE10 pairs by the number of double hexamers added to the number of sCMGE10 pairs—that is, (sCMGE / 2) / (DH + sCMGE / 2). This is because two sCMGE complexes would result from any given double hexamer. When sCMGE, dCMGE and double hexamer 2D classes were found, conversion efficiency for sCMGEs was calculated by dividing the number of sCMGE10 pairs by the number of double hexamers added to the number of sCMGE10 pairs added to the number of dCMGEs—that is, (sCMGE / 2) / (DH + sCMGE / 2 + dCMGE).

### Graphene oxide grid preparation

UltrAuFoil R1.2/1.3 grids (Quantifoil Micro Tools) were freshly glow discharged for 5 min at 25 mA using a glow discharge unit (GloQube Plus, Quorum). Graphene oxide dispersion (2 mg ml⁻¹, Sigma-Aldrich) was diluted by adding 80 µl of water to 10 µl of graphene oxide dispersion. Diluted dispersion was spun down at 500g for 1 min. The top layer of dispersion was transferred to a new tube, avoiding aggregates. For the grid preparation, 4 µl of diluted graphene oxide dispersion was applied to the freshly glow discharged UltrAuFoil R1.2/1.3 grid and incubated for 2 min. Both sides of the grid were blotted. Droplets of water, 20 µl each, were picked up with the grid and blotted, twice for the front and once for the back of the grid. Grids were then dried upside down.

### Cryo-EM sample preparation and data collection

Cryo-EM sample preparation was carried out by adapting previously published protocols[11]. CMG assembly and activation reactions (reconstituted as described in in vitro CMG assembly and activation on short MH-flanked origins) were frozen on UltrAuFoil R1.2/1.3 grids (Quantifoil Micro Tools) with a freshly prepared graphene oxide layer. All grids were prepared as detailed above before freezing. Samples were prepared by applying 4 µl of undiluted CMG assembly and activation reaction on grids, incubated for 2 min at 25 °C in 90% humidity. Excess sample was subsequently blotted away for 4.5 s or 5.0 s, and grids were plunge frozen in liquid ethane using a Vitrobot Mark IV (FEI Thermo Fisher Scientific).

Data collection was performed on an in-house Thermo Fisher Scientific Titan Krios transmission electron microscope operated at 300 kV equipped with a Gatan K2 direct electron detector camera and a GIF Quantum energy filter (Gatan). Images were collected automatically using EPU software (Thermo Fisher Scientific) in counting mode with a physical pixel size of 1.08 Å per pixel, with a total electron dose of 51.4 electrons per Å² during a total exposure time of 10 s, dose fractionated into 32 movie frames (Table 1). We used a slit width of 20 eV on the energy filter and a defocus range of −1.8 µm to −3.3 µm. A total of 71,117 micrographs were collected from two separate sessions.

### Cryo-EM image processing

Data processing was carried out in RELION 3.1 (ref. 50) or cryoSPARC version 4.0.3 (ref. 53) (Extended Data Fig. 3). Correction for movie drift and dose weighting was performed using MotionCorr2 (ref. 54). CTF parameters were estimated for the drift-corrected micrographs using CtfFind4 (ref. 55) in RELION 3.1. First, dataset 1, which was collected first, was processed separately and, at a later stage, combined with dataset 2. For the first dataset, particles were manually picked from 2,000 micrographs using cryoSPARC version 4.0.3 (ref. 53). These particles were extracted with a box size of 420² pixels for 2D. Starting with this subset of particles across the entire defocus range, a Topaz model[51] was trained iteratively to improve particle picking. In RELION 3.1, particles were picked from both datasets (first and second collection, total 71,116 micrographs) with a select threshold of 0.

The two datasets were combined, and a total of 2,016,248 particles were picked, binned by 2 and extracted with a box size of 180² pixels. Picked particles were subjected to 2D classification to remove remaining smaller particles and contaminants. With the remaining particles, we carried out a 3D classification with two or three subclasses, angular sampling of 7.5°, regularization parameter T of 4 using a low-pass filtered initial model from previous ab initio and processing steps on dataset 1 of CMGE complexes (Extended Data Fig. 3). At the same stage, all particles were subjected to 2D classification to select for high-resolution 2D classes. Selection of 2D and 3D classes were combined, yielding 244,182 particles. These particles were un-binned to then perform another round of 3D classification with two subclasses, angular sampling of 7.5°, regularization parameter T of 8 using a low-pass filtered model from previous 3D classification. The resulting 172,552 CMGE particles were refined to yield maps with resolutions of 4.4 Å.

These particles were subjected to several rounds of CTF refinement and two rounds of Bayesian polishing. After this, CTF-refined and polished particles were refined with local searches with a mask encompassing the entire CMGE density to 3.7 Å resolution, the 'consensus' structure (Extended Data Fig. 2 and Table 1).

From 3D classifications, we found that some 3D classes contain double-stranded DNA (dsDNA) inside the central channel of CMG toward the C-terminus of CMG. 3D classification, however, gave poor particle separation of CMGE particles on ssDNA and CMGE particles with dsDNA in the C-terminal region. To better separate these two particle sets, we used signal subtraction in combination with reference-free 2D classification without alignment in RELION 3.1 (refs. 8,50). Using this approach, we separated side views; a subset of 13,174 particles was selected as being threaded on only ssDNA, and a subset of 23,337 particles was selected with dsDNA in the C-terminal region. All signal-subtracted particles were then reverted. To separate top and bottom views of CMGE that contain

ssDNA from CMGE that contain dsDNA, we carried out a 3D classification of the remaining particles and particle side views with dsDNA. The 3D classification was performed with 20 subclasses, angular sampling of 7.5° and regularization parameter T of 16, and resolution was limited to E = 12, using a low-pass filtered model from previous refinement. 3D classes were categorised into CMGEs with either ssDNA or dsDNA in C-terminus of the central channel. Particles from 2D classified side views mentioned above were also added accordingly to the particle sets of CMGE on ssDNA or dsDNA. These two particle sets were refined using local searches to 3.9-Å resolution for ssDNA-bound CMGE complexes and 3.9-Å resolution for particles with dsDNA in the C-terminal region (Table 1).

All refinements were performed using fully independent data half-sets, and resolutions are reported based on the Fourier shell correlation (FSC) = 0.143 criterion (Extended Data Fig. 2). FSCs were calculated with a soft mask. Maps were corrected for the modulation transfer function of the detector and sharpened by applying a negative B-factor as determined by the post-processing function of RELION or using a higher B-factor to prevent overfitting. PyEM (https://github.com/asarnow/pyem) and bsfot (https://cbiit.github.io/Bsoft/) were used for format conversion.

### Model building and refinement

To generate a suitable starting point for model building, deposited coordinates for CMGE (Protein Data Bank (PDB) 7QHS) were split into MCM subdomains, each of which was rigid body docked into a refined consensus volume in UCSF Chimera[56], along with chains corresponding to the GINS subcomplex and Cdc45. DNA polymerase epsilon catalytic subunit A coordinates AA 1551–1585 were obtained from PDB 7PMK. Refined maps were converted to MTZ format using the mrc2mtz module. The coordinates were then rebuilt and extended in Coot[57] according to the density of blurred and sharpened outputs from mrc2mtz. To address steric clashes and geometric outliers, the model was further adjusted using ISOLDE[58], resulting in a base set of coordinates. As data processing revealed an ssDNA-bound and a nexus-bound state within the consensus particle set, maps for these subclassified volumes were also produced for model building. The base coordinates were then docked into these maps and adjusted according to the density. Each set of coordinates was then refined separately in PHENIX[59]. Atomic model geometries were evaluated using MolProbity webserver[60]. Mcm10 docking was performed using UCSF Chimera[56] and locally adjusted using ISOLDE[58].

### Map and model visualization

Maps and all model illustrations were visualized and prepared using Chimera or ChimeraX[56,61].

### Reporting summary

Further information on research design is available in the Nature Portfolio Reporting Summary linked to this article.

### Data availability

The authors declare that the data supporting the findings of this study are available within the paper and its Supplementary Information files. Cryo-EM density maps of the consensus CMGE10 complex have been deposited in the Electron Microscopy Data Bank (EMDB) under accession code EMD-17459. Cryo-EM density maps of the CMGE on ssDNA (ssDNA-sCMGE10) have been deposited in the EMDB under accession code EMD-17458, CMGE with dsDNA in C-terminus (nexus-sCMGE10) under accession code EMD-17449 and the binned map with Polε signal subtraction, nexus-sCMG10, under accession code EMD-17460. Atomic coordinates have been deposited in the Protein Data Bank with accession codes 8P63, 8P62 and 8P5E for the consensus structure, ssDNA-sCMGE and nexus-sCMGE, respectively. The integrative atomic model of the

sCMGE10 complex combines structural and cross-linking mass spectrometry data and can be provided by the corresponding author upon reasonable request. Source data are provided with this paper.

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

## Acknowledgements

We would like to thank M. Douglas and the members of the Costa laboratory for useful discussions. We acknowledge the Crick Structural Biology STP for computational support (A. Purkiss and P. Walker); yeast cultures (N. Patel, A. Alidoust and D. Patel) and cryo-EM support (D. Benton). This work was funded jointly by the Wellcome Trust, the Medical Research Council and Cancer Research UK at the Francis Crick Institute (CC2009 and CC2002). A.C. receives funding from the European Research Council (ERC) under the European Union's Horizon 2020 Research and Innovation Program (grant agreement no. 820102). S.S.H. is the recipient of a Human Frontier Science Program award (LT000834/2020-L) and a European Molecular Biology

Organization (EMBO) long-term fellowship award (ALTF 962-2019). J.S.L. and M.H.G. are the recipients of EMBO long-term fellowship awards (ALTF 211-2020 and 34-2021) and Marie Skłodowska-Curie Actions fellowship awards (101018683, DNA-Rep-EM, and 101025825, MechHelicaseActiv8on). T.P. and G.P. are the recipients of Boehringer Ingelheim Fonds PhD fellowships. This work was also funded by a Wellcome Trust Senior Investigator Award (219527/Z/19/Z) and an ERC Advanced Grant (101020432-MeChroRep) to J.F.X.D.

## Author contributions

S.S.H. and A.C. conceived the study. S.S.H. performed DNA binding, helicase loading and activation assays, negative-stain electron microscopy, cryo-electron microscopy and image processing. T.P. conducted biochemical experiments for paper revisions. T.P. and A.C. helped with negative-stain data collection, and J.F.G., J.S.L. and G.P. helped with negative-stain data processing. Atomic model building and refinement were performed by S.S.H. and O.W. S.S.H. prepared all reagents for structural analysis, with the help of J.S.L. S.S.H., M.H.G., A.W.M. and J.F.G. prepared reagents for DNA replication assays, which were conducted by M.H.G. and A.W.M., under the supervision of J.F.X.D. M.H.G. also performed pulse-chase and DNA unwinding assays. A.N. provided critical support with image acquisition on the Krios microscope. S.S.H. and A.C. wrote the paper, with input from the other authors. A.C. supervised the study.

## Funding

## Competing interests

The authors declare no competing interests.

## Additional information

**Extended data** is available for this paper at https://doi.org/10.1038/s41594-024-01280-z.

**Correspondence and requests for materials** should be addressed to Alessandro Costa.

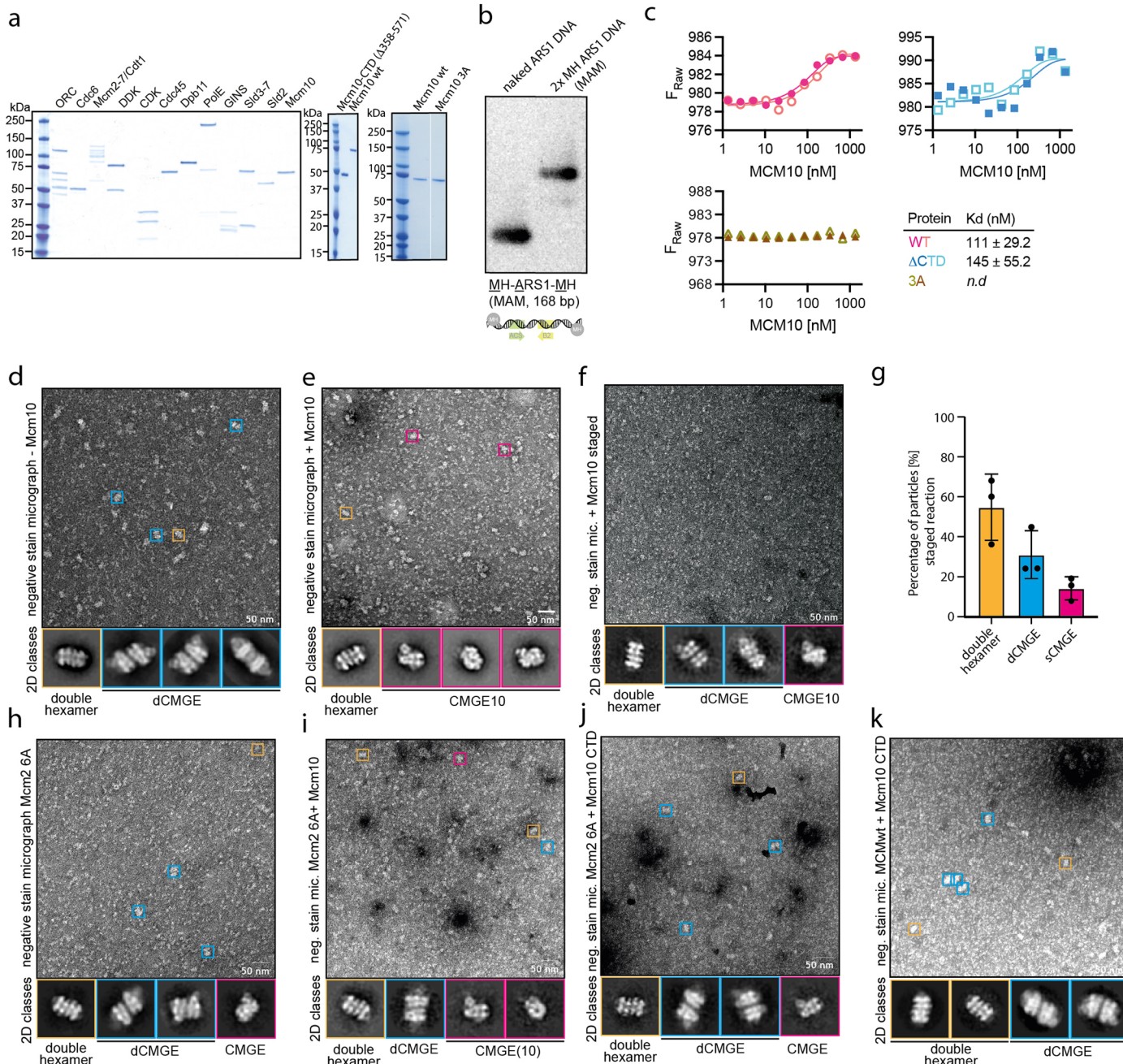

**Extended Data Fig. 1 | Mcm10 dependent origin unwinding visualised by negative stain electron microscopy. (a)** Purified yeast proteins and Mcm10 mutants used in the origin unwinding reaction and MST assay. **(b)** ARS1 origin substrate blocked at two ends with covalently linked MH molecules. **(c)** Wild type Mcm10 and C-terminal truncation of Mcm10 bind single-stranded DNA with nanomolar affinity. Mcm10 3A (OB-fold) mutant does not retain binding affinity to single-stranded DNA. Duplicate plotted. **(d)** Micrograph and 2D averages of a negatively stained preparation show double CMGE assembled when Mcm10 is omitted. **(e)** Micrograph and 2D averages of a negatively stained preparation show that addition of Mcm10 yields split CMGE10 particles, with no double CMGE observed. **(f)** Micrograph and 2D averages of a negatively stained staged reaction where Mcm10 is added after double CMGE formation. Splitting of double CMGEs into sCMGE10 is observed, although double CMGEs are still observed in the preparation. **(g)** Quantification of double hexamers, double CMGEs and

sCMGE10s for staged reaction. Experiment performed three times. Error bars, mean ± s.d. **(h)** Micrograph and 2D averages of a negatively stained double CMGE assembly reaction using the Mcm2 6A mutant shows formation of double as well as sCMGE, in the absence of Mcm10. **(i)** Micrograph and 2D averages of a negatively stained CMGE assembly reaction using the Mcm2 6A mutant in the presence of Mcm10 shows Mcm10 binding to N-terminal MCM in a single CMGE complex. **(j)** Micrograph and 2D averages of a negatively stained staged CMGE assembly reaction using the Mcm2 6A mutant and a C-terminal truncation of Mcm10 shows sCMGEs particles that are not decorated with Mcm10 at MCM N-terminal domain. **(k)** Micrograph and 2D averages of a negatively stained staged CMGE assembly reaction using the C-terminal truncation of Mcm10 shows double CMGEs but not sCMGE10 complex formation. Micrographs and averages shown derive from experiments performed in triplicates. 2D averages obtained for each experiment derive from processing ~50 to ~200 micrographs.

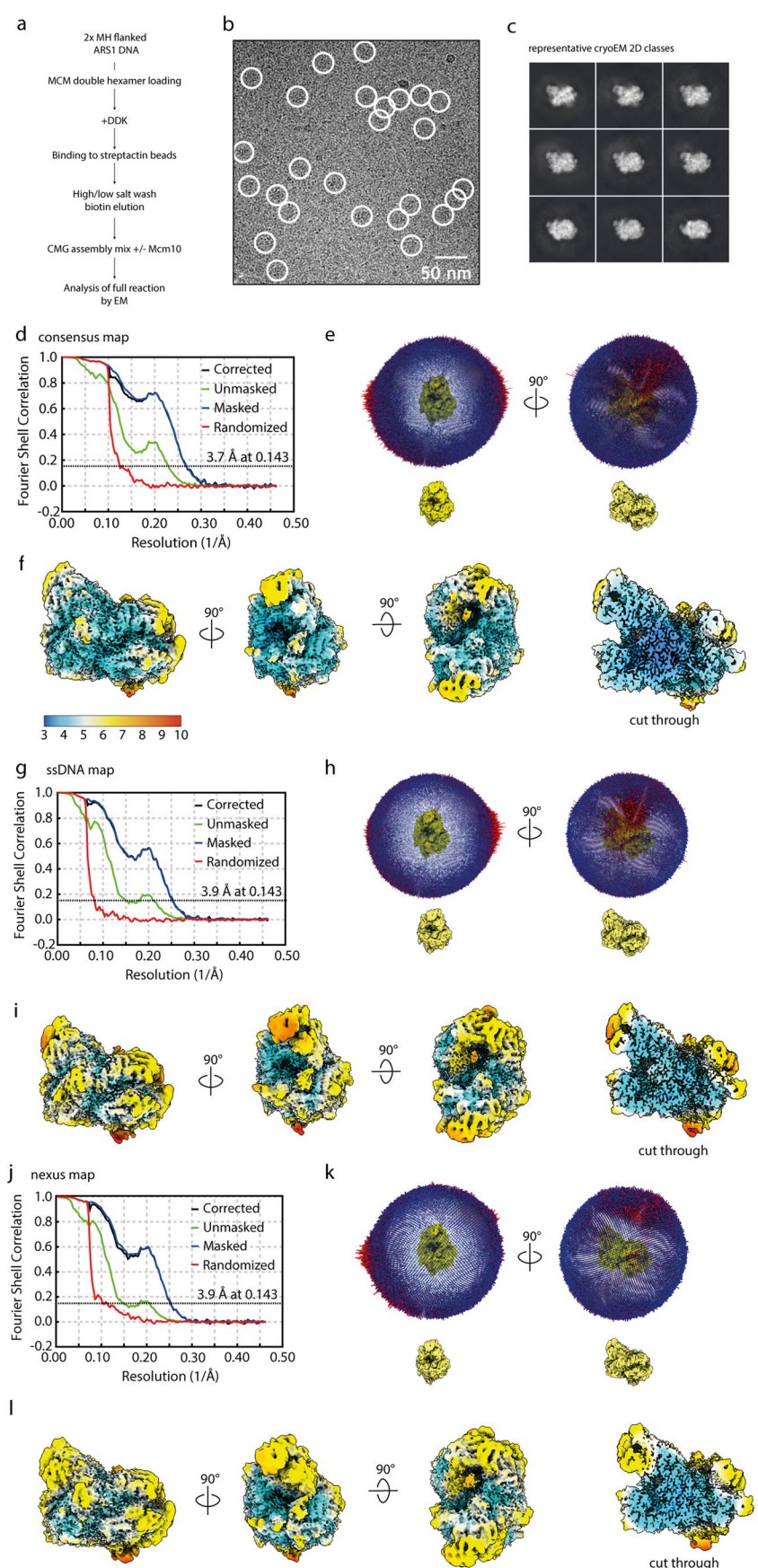

**Extended Data Fig. 2 | See next page for caption.**

**Extended Data Fig. 2 | Cryo-EM analysis of sCMGE10. (a)** Sample preparation pipeline. **(b)** Representative sum of aligned movie frames. **(c)** Representative 2D averages. **(d)** Fourier shell correlation plot for consensus map. **(e)** Angular distribution for consensus map. **(f)** Cryo-EM map filtered and coloured according to local resolution for consensus map. **(g)** Fourier shell correlation plot for ssDNA map. **(h)** Angular distribution for ssDNA-CMGE10 map. **(i)** Cryo-EM map filtered and coloured according to local resolution for ssDNA-CMGE10 map. **(j)** Fourier shell correlation plot for nexus-CMGE10 map. **(k)** Angular distribution for nexus-CMGE10 map. **(l)** Cryo-EM map filtered and coloured according to local resolution for nexus-CMGE10 map.

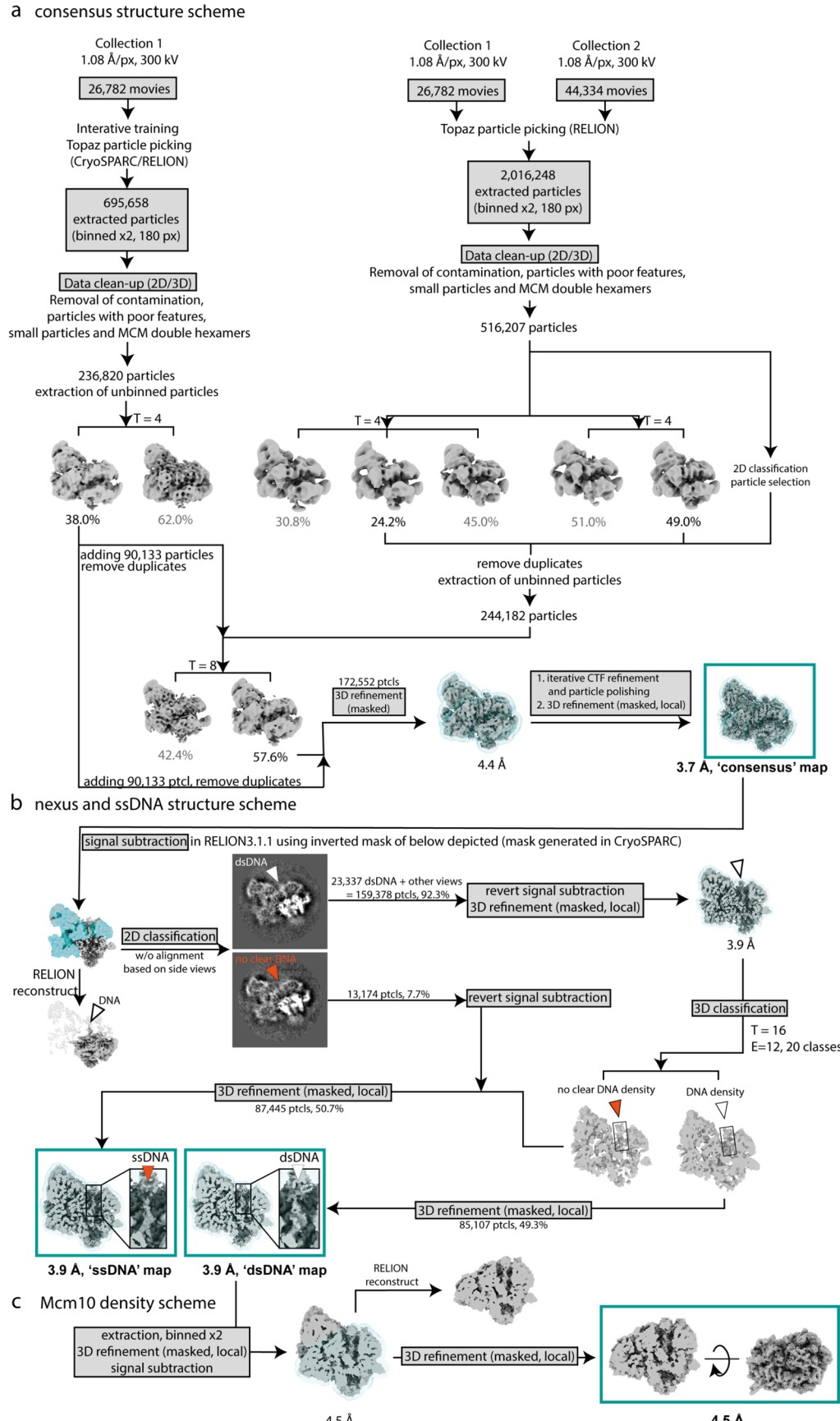

**Extended Data Fig. 3 | Cryo-EM image processing pipeline. (a)** Procedure used to obtain a consensus sCMGE10 structure. **(b)** Procedure used to separate ssDNA-sCMGE10 complexes from nexus-sCMGE10 complexes. **(c)** Local refinement procedure used to obtain interpretable Mcm10 density.

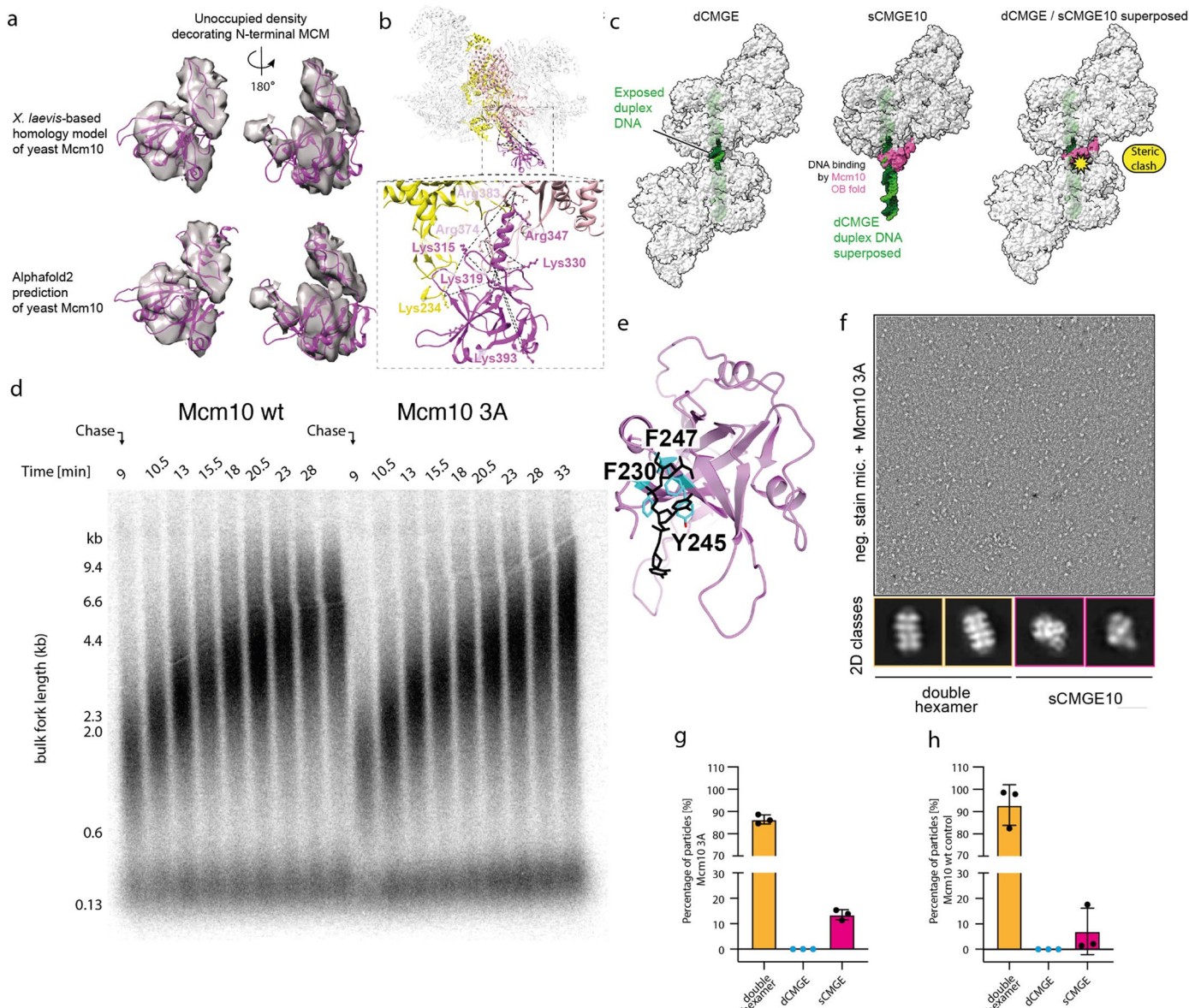

**Extended Data Fig. 4 | Docking validation of Mcm10 OB-fold and ZnF domains. (a)** Homology model for *S. cerevisiae* Mcm10 based on the *X. laevis* crystal structure (PDB entry 3H15) and Alphafold2 prediction docked into unoccupied density decorating N-terminal MCM in s nexus-sCMGE10. The highest correlation docking solution was the same for the two atomic models. **(b)** The highest correlation solutions obtained in both docking experiments best satisfy the distance constraints imposed by the disuccinimidyl suberate crosslinker used in a published mass-spectrometry study[15]. 100% of crosslinks within the modelled region feature inter-beta carbon distances below 50 Å and 63% of crosslinks are shorter than 35 Å. For the second-best docking solution (not displayed), only 82% of the measured crosslinks are shorter than 50 Å and 55% shorter than 35 Å. **(c)** Mcm10 OB fold/ZnF domains binding to N-terminal MCM in a sCMGE clashes with the second CMGE in dCMGE. Hence, Mcm10 binding to

the N-terminal MCM makes dCMGE splitting irreversible. **(d)** Pulse-chase DNA replication assay reveals that the Mcm10 3A mutant has no detectable defect in DNA replication and no obvious change in fork rate. **(e)** Atomic modelling combining the co-crystal structure of *Xenopus laevis* Mcm10 and single stranded DNA with the yeast AlphaFold model of *S. cerevisiae* Mcm10 allowed the identification of three residues responsible for single-stranded DNA binding in the yeast proteins. This information was used to generate the Mcm10 3A mutant. **(f)** Negative stain electron micrograph and 2D averages of the Mcm10 3A mutant. **(g-h)** Quantification indicates that this Mcm10 variant has no loss in efficiency of dCMGE splitting, revealing that single-stranded DNA binding by Mcm10 does not have a role in initiation. Experiment performed three times. Error bars, mean ± s.d.

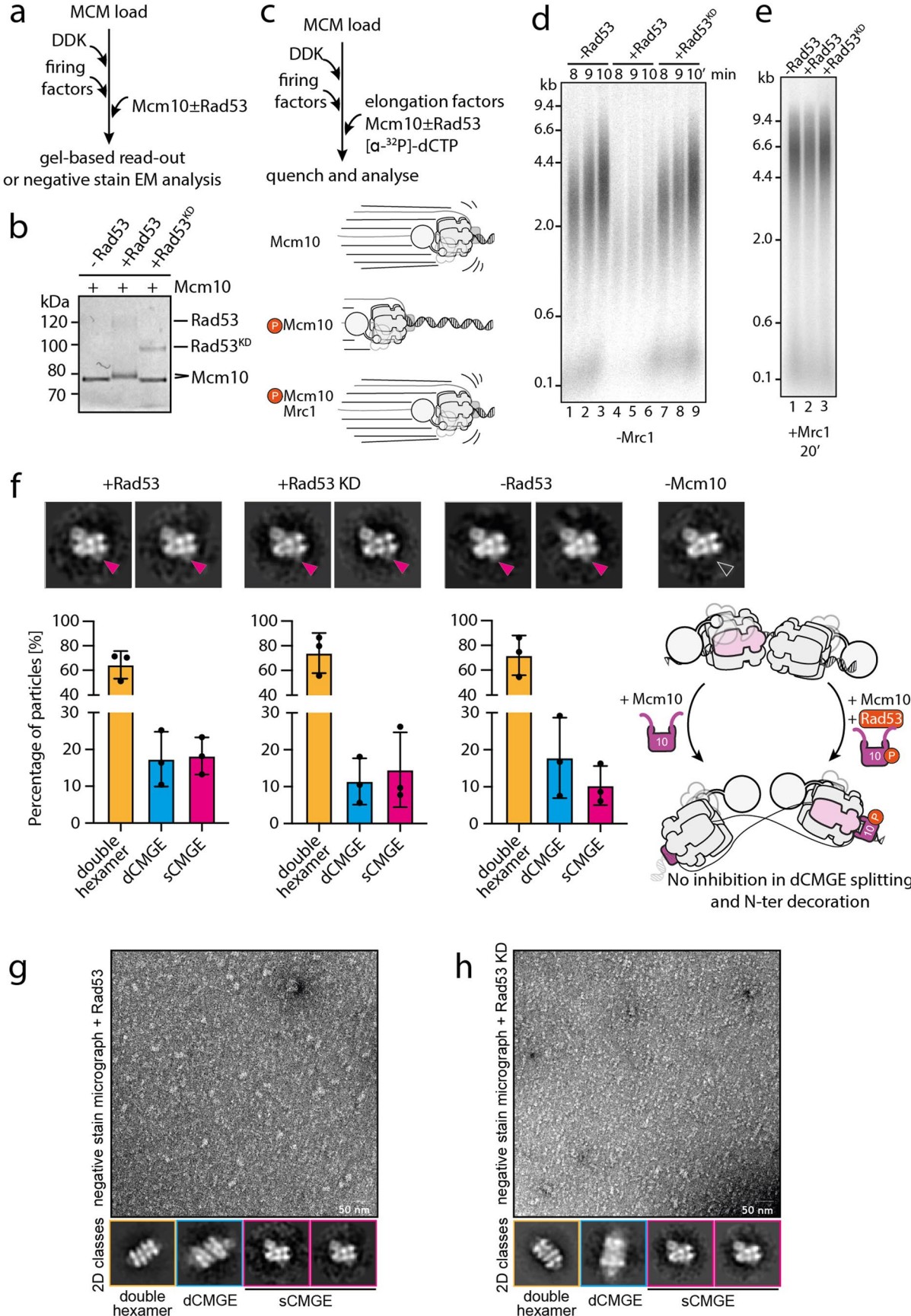

**Extended Data Fig. 5 | See next page for caption.**

**Extended Data Fig. 5 | Rad53 modulates fork rate, not helicase splitting.**
(a) Workflow for sCMGE10 assembly with Rad53-phosporylated Mcm10.
(b) A phosphorylation-dependent shift in Mcm10 migration can be observed by SDS-PAGE analysis when using wild type but not catalytically-dead Rad53.
(c) Workflow for DNA replication reconstituted *in vitro* with or without Mcm10 phosphorylation by Rad53. (d) Wild type but not catalytically-dead Rad53 severely affects DNA replication. (e) Addition of Mrc1 after 20 minutes yields DNA replication for the phosphorylated Mcm10 reaction, which is comparable to the unphosphorylated Mcm10 reaction, indicating that Rad53 phosphorylation of Mcm10 affects fork progression speed and not replication initiation. (f) Rad53 phosphorylation does not affect N-terminal MCM decoration by Mcm10, nor the efficiency of dCMGE splitting in a staged reaction. (g) Negative stain micrographs and 2D averages of sCMGE10 assembled using Rad53-pre-phosphorylated Mcm10 shows that N-terminal MCM decoration by Mcm10 in sCMGE10 is not affected. (h) Control experiment showing sCMGE10 formation using Mcm10 pre-incubated with catalytically-dead Rad53.

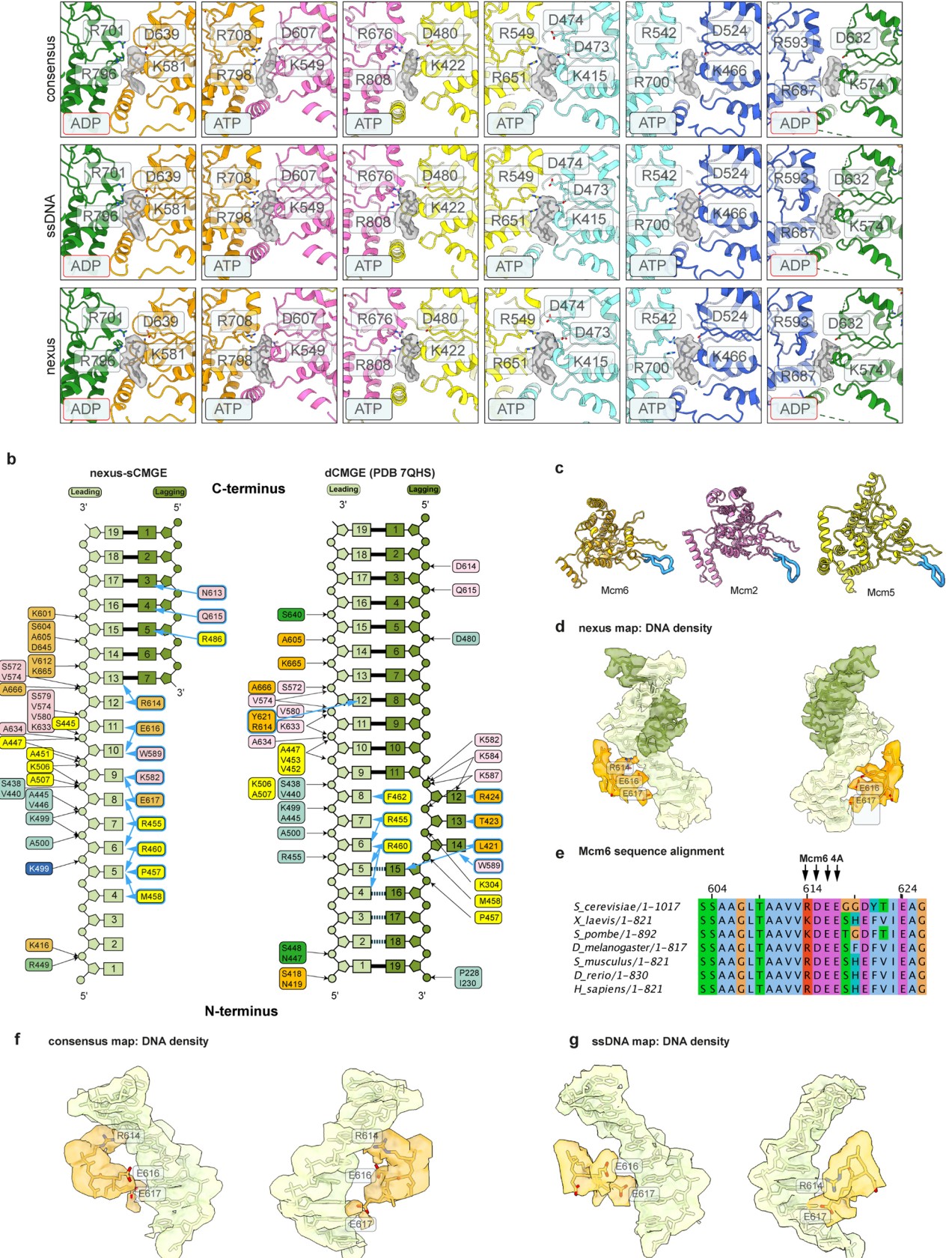

**Extended Data Fig. 6 | Details of cryo-EM densities and DNA interactions.**
(a) Cryo-EM density for nucleotide ATP and ADP for consensus, ssDNA and nexus map. (b) DNA interacting elements in nexus-sCMGE10 in comparison to DNA interaction observed in dCMGE (PDB 7QHS). (c) Helix 2 insert (h2i) pore loop of Mcm2, Mcm5 and Mcm6. (d) Cryo-EM density for the duplex-single stranded DNA nexus, engaged by residues R614, E616 and E617. (e) Sequence alignment showing that these residues are highly conserved across eukaryotes. (f) Cryo-EM density for the consensus map. (g) Cryo-EM density for the ssDNA map.

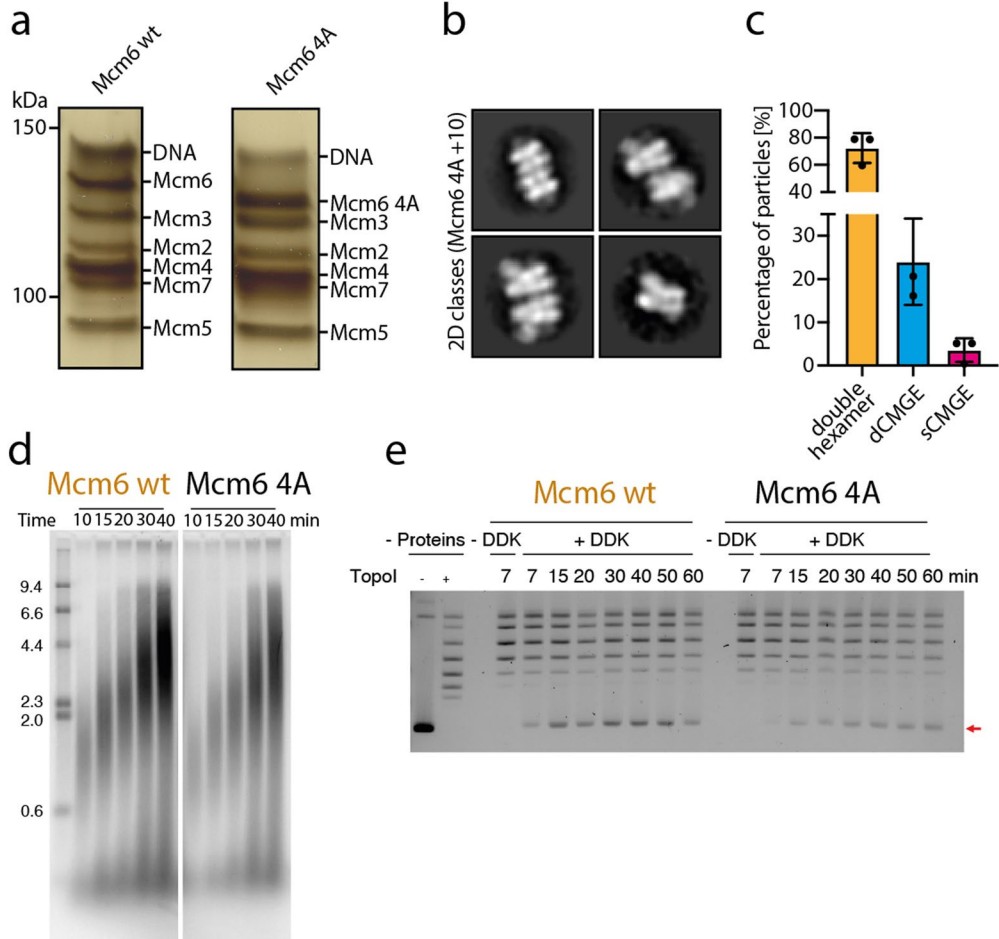

**Extended Data Fig. 7 | Effects of mutation and modification on dCMGE10 splitting. (a)** Silver stained SDS-PAGE gel showing that Mcm6 4A MCM can be loaded onto a roadblocked origin DNA to wild type levels. **(b)** 2D averages showing that Mcm6 4A supports dCMGE and sCMGE10 formation. **(c)** Quantification shows that sCMGE10 formation is negatively affected by the Mcm6 4A mutation. Experiment performed three times. Error bars, mean ± s.d.

**(d)** Time course experiment shows that DNA replication with the Mcm6 4A variant is partially impaired. This is a repeat of the experiment shown in Fig. 6c. **(e)** Time-course plasmid-based DNA unwinding assay similar to that shown in Fig. 6d and shows the same trend, despite an increase of ORC concentration from 10 to 30 nM.

# Reporting Summary

## Statistics

For all statistical analyses, confirm that the following items are present in the figure legend, table legend, main text, or Methods section.

| n/a | Confirmed | |
|---|---|---|
| ☐ | ☒ | The exact sample size (*n*) for each experimental group/condition, given as a discrete number and unit of measurement |
| ☐ | ☒ | A statement on whether measurements were taken from distinct samples or whether the same sample was measured repeatedly |
| ☒ | ☐ | The statistical test(s) used AND whether they are one- or two-sided<br>*Only common tests should be described solely by name; describe more complex techniques in the Methods section.* |
| ☒ | ☐ | A description of all covariates tested |
| ☒ | ☐ | A description of any assumptions or corrections, such as tests of normality and adjustment for multiple comparisons |
| ☐ | ☒ | A full description of the statistical parameters including central tendency (e.g. means) or other basic estimates (e.g. regression coefficient) AND variation (e.g. standard deviation) or associated estimates of uncertainty (e.g. confidence intervals) |
| ☐ | ☒ | For null hypothesis testing, the test statistic (e.g. *F*, *t*, *r*) with confidence intervals, effect sizes, degrees of freedom and *P* value noted<br>*Give P values as exact values whenever suitable.* |
| ☒ | ☐ | For Bayesian analysis, information on the choice of priors and Markov chain Monte Carlo settings |
| ☒ | ☐ | For hierarchical and complex designs, identification of the appropriate level for tests and full reporting of outcomes |
| ☒ | ☐ | Estimates of effect sizes (e.g. Cohen's *d*, Pearson's *r*), indicating how they were calculated |

*Our web collection on statistics for biologists contains articles on many of the points above.*

## Software and code

Policy information about availability of computer code

| Data collection | Gatan DigitalMicrograph, ThermoFisher EPU v2.9 and Monolith NT.115 |
|---|---|
| Data analysis | Topaz v0.2.4, MotionCor2, Ctffind v4.1.13, RELION v3.1, cryoSPARC 4.0, pyem v0.4, bsoft v2.0.4, UCSF Chimera v1.14, ChimeraX-1.3, COOT v0.9-pre, ISOLDE, Phenix v1.20.1, MolProbity web sever, ImageJ2/Fiji v2.3.0, GraphPad Prism v9.4.1 |

For manuscripts utilizing custom algorithms or software that are central to the research but not yet described in published literature, software must be made available to editors and reviewers. We strongly encourage code deposition in a community repository (e.g. GitHub). See the Nature Portfolio guidelines for submitting code & software for further information.

## Data

Policy information about availability of data

All manuscripts must include a data availability statement. This statement should provide the following information, where applicable:
- Accession codes, unique identifiers, or web links for publicly available datasets
- A description of any restrictions on data availability
- For clinical datasets or third party data, please ensure that the statement adheres to our policy

Cryo-EM density consensus maps of the CMGE10 complex has been deposited in the Electron Microscopy Data Bank (EMDB) under the accession number EMD-17459. Cryo-EM density map of ssDNA CMGE10 map has been deposited in the EMDB under the accession number EMD-17458. Cryo-EM density map of nexus-CMGE10 map has been deposited in the EMDB under the accession number EMD-17449. Atomic coordinates have been deposited in the Protein Data Bank (PDB) with the accession numbers 8P63 (consensus CMGE), 8P62 (ssDNA-CMGE) and 8P5E (nexus-CMGE on dsDNA).

# Field-specific reporting

Please select the one below that is the best fit for your research. If you are not sure, read the appropriate sections before making your selection.

☒ Life sciences　　☐ Behavioural & social sciences　　☐ Ecological, evolutionary & environmental sciences

For a reference copy of the document with all sections, see nature.com/documents/nr-reporting-summary-flat.pdf

# Life sciences study design

All studies must disclose on these points even when the disclosure is negative.

| | |
|---|---|
| Sample size | In our negative stain EM experiments, we imaged Mcm10-dependent CMG activation, yielding different reaction intermediates. To isolate CMGs, we usually collected 50-300 micrographs per condition. The sample size was sufficient to either allow 2D classification or comparative analysis between MCM or Mcm10 mutants. All these experiments were performed three times.<br><br>To obtain high-resolution structure of the ssDNA- or dsDNA-bound CMGE in complex with Mcm10, ~71.1 K micrographs were collected from two independent grids made from the same CMG activation reaction. This sample size was appropriate and sufficient to allow model building or comparative analysis.<br><br>No statistical methods were used to predetermine sample size. |
| Data exclusions | Negative stain and cryo-EM micrographs with poor staining or ice contamination, respectively, were excluded. Picked particles that did not align to a distinct class in 2D and 3D (cryo-EM only) were excluded from further analysis. |
| Replication | The cryo-EM dataset of Mcm10-dependent CMG activation reaction comprised of a single reaction and two datasets, collected on two independent grids. CMGE10 complex formation in negative stain EM experiments was found to be reproducible across multiple independent sample preparations. Details of the number of experimental repeats have been acknowledged in the relevant figure legends. Details of the number of experimental repeats have been acknowledged in the relevant figure legends. All attempts at data replication were successful. |
| Randomization | For calculation of the resolution of the cryo-EM reconstructions, Fourier shell correlations were calculated using independent halves of the complete datasets, into which the component particles were segregated randomly. |
| Blinding | Blinding is not relevant for a single particle electron microscopy study such as this. No risk of bias identifiable. |

# Reporting for specific materials, systems and methods

We require information from authors about some types of materials, experimental systems and methods used in many studies. Here, indicate whether each material, system or method listed is relevant to your study. If you are not sure if a list item applies to your research, read the appropriate section before selecting a response.

## Materials & experimental systems

| n/a | Involved in the study |
|---|---|
| ☒ | ☐ Antibodies |
| ☐ | ☒ Eukaryotic cell lines |
| ☒ | ☐ Palaeontology and archaeology |
| ☒ | ☐ Animals and other organisms |
| ☒ | ☐ Human research participants |
| ☒ | ☐ Clinical data |
| ☒ | ☐ Dual use research of concern |

## Methods

| n/a | Involved in the study |
|---|---|
| ☒ | ☐ ChIP-seq |
| ☒ | ☐ Flow cytometry |
| ☒ | ☐ MRI-based neuroimaging |

## Eukaryotic cell lines

Policy information about cell lines

| | |
|---|---|
| Cell line source(s) | S. cerevisiae overexpression strains for CMG assembly and DNA replication proteins have previously been described in multiple studies across several publications. For clarity to the potential readers and reviewers we have included extensive details in extended data table X. |
| Authentication | S. cerevisiae overexpression strains were checked for correct plasmid integration by PCR amplification from extracted genomic DNA. |
| Mycoplasma contamination | S. cerevisiae overexpression strains were not tested for mycoplasma contamination. |

Commonly misidentified lines
(See ICLAC register)

No commonly misidentified cell lines were used in this study.

