## [Peer Review File · Nature Structural & Molecular Biology]

Peer Review Information

Manuscript Title: Unwinding of a eukaryotic origin of replication visualised by cryo-EM

Corresponding author name(s): Alessandro Costa

Editorial Notes:

Transferred manuscripts This manuscript has been previously reviewed at another journal that is not operating a transparent peer review scheme. This document only contains reviewer comments, rebuttal and decision letters for versions considered at Nature Structural & Molecular Biology

Reviewer Comments & Decisions:

Decision Letter, initial version:
--

Message: Our ref: NSMB-A48092-T

1st Feb 2024

Dear Dr. Costa,

Thank you for submitting your revised manuscript "Unwinding of a eukaryotic origin of replication visualised by cryo-EM" (NSMB-A48092-T). It has now been seen by the original referees and their comments are below. The reviewers find that the paper has further improved in revision, and therefore we will be happy to accept it in principle in Nature Structural & Molecular Biology, pending minor revisions to satisfy the referees' final requests and to comply with our editorial and formatting guidelines.

We are now performing detailed checks on your paper and will send you a checklist detailing our editorial and formatting requirements in about two weeks. Please do not upload the final materials and make any revisions until you receive this additional information from us.

To facilitate our work at this stage, it is important that we have a copy of the main text as a word file. If you could please send along a word version of this file as soon as possible, we would greatly appreciate it; please make sure to copy the NSMB account (cc'ed above).

Thank you again for your interest in Nature Structural & Molecular Biology and congratulations on the acceptance in principle of your manuscript. Please do not hesitate to contact me if you have any questions.

Sincerely,

Dimitris Typas
Associate Editor
Nature Structural & Molecular Biology
ORCID: 0000-0002-8737-1319

Reviewer #1 (Remarks to the Author):

In this revision, the authors have added new experimental data and revised the manuscript text to successfully address my main concerns. However, the authors should carefully check their manuscript and figures to ensure they are correctly labelled and referenced. A few inconsistencies and issues that remain and should be addressed are:

1. Lines 234-239: The authors seem to reference the wrong Extended data figure panels because the results don't agree with the statements in the text. Why is there CMGE10 in Extended data fig. 1j if the Mcm10-CTD is missing?
2. Fig. 6c: If the authors cannot include error bars, the results of the replicate experiments (the quantification) should be shown considering the difference in DNA replication seen is extremely small.
3. Fig. 6d: this figure would benefit from quantification of the indicated bands with replicate experiments (red arrow).
4. Fig. 6f is not included in the figure legend or referenced in the text.
5. Extended data fig. 7b and c: These reactions seem to lack Mcm10. How can a sCMGE10 complex (defined as containing Mcm10 by the authors) be formed?
6. Extended data fig. 6a (now 7a): Negative controls are not included in this figure contrary to the authors' statement in the rebuttal.

Reviewer #2 (Remarks to the Author):

The authors have satisfactorily addressed all comments raised in the prior round of review. The only potential item of note is that in one of the responses ("Density for DNA substrates and ATPase sites needs to be clearly shown..."), what is listed as Extended Data Figure 5 should now be Extended Data Figure 6? Publication in NSMB is recommended.

Reviewer #3 (Remarks to the Author):

The revised manuscript by Costa, Diffley, and colleagues describes a structural and biochemical analysis of Mcm10-activated DNA unwinding. They demonstrate how Mcm10 contributes to stable separation of the two activated helicases during initiation and provide insights into the initial DNA unwinding steps.

The revisions include new biochemical experiments that substantially expand the conclusions regarding the function of the N- and C-terminus of Mcm10 as well as interesting new data concerning the role of Mcm10 and Mrc1 during the elongation stages of replication. In addition the authors have added new data addressing the role of the ssDNA domain of Mcm10 in the function of this proteins during initiation and elongation. Finally, the revised manuscript does a much better job of integrating the Rad53 findings into the main theme of the paper.

The revised paper is much improved and appropriate for publication in NSMB. It provides important new information about Mcm10 function and the nature of initial DNA unwinding during helicase activation at eukaryotic origins. Although a manuscript from O'Donnell and colleagues addressing the same step of initiation has been published in the interim, those studies are highly artificial and the studies presented here, while in agreement with the proposed mechanism, are performed in the context of a true initiation reaction.

Author Rebuttal to Initial comments

Reply to the referees

We would like to thank the referees for their positive assessment of our revised work. Outstanding issues have been addressed in our revised manuscript as detailed below.

Referee #1:

1. Lines 234-239: The authors seem to reference the wrong Extended data figure panels because the results don't agree with the statements in the text. Why is there CMGE10 in Extended data fig. 1j if the Mcm10-CTD is missing?

Thank you for spotting this issue! That was a critical oversight. The CMGE10 label should have been CMGE instead. In fact, no Mcm10 is visible. Now corrected.

2. Fig. 6c: If the authors cannot include error bars, the results of the replicate experiments (the quantification) should be shown considering the difference in DNA replication seen is extremely small.

We now show a repeat of the experiment in Extended Data Figure 7d.

3. Fig. 6d: this figure would benefit from quantification of the indicated bands with replicate experiments (red arrow).

We feel that including another DNA unwinding experiment showing the same trend observed in Figure 6d is most informative. This is shown in Extended Data Figure 7e.

4. Fig. 6f is not included in the figure legend or referenced in the text.

Thank you for spotting this. Now corrected.

5. Extended data fig. 7b and c: These reactions seem to lack Mcm10. How can a sCMGE10 complex (defined as containing Mcm10 by the authors) be formed?

The reactions contain Mcm10 but the – "(Mcm6 4A – Mcm10)" label was very confusing. We now changed it to "(Mcm6 4A + Mcm10)".

6. Extended data fig. 6a (now 7a): Negative controls are not included in this figure contrary to the authors' statement in the rebuttal.

That is correct apologies.

Referee #2:

The only potential item of note is that in one of the responses ("Density for DNA substrates and ATPase sites needs to be clearly shown..."), what is listed as Extended Data Figure 5 should now be Extended Data Figure 6?

That is correct. Thank you for noting this issue. No further action taken as the remark pertains to the previous reply to referees only.

Final Decision Letter:

Message: 19th Mar 2024

Dear Dr. Costa,

We are now happy to accept your revised paper "Unwinding of a eukaryotic origin of replication visualised by cryo-EM" for publication as an Article in Nature Structural & Molecular Biology.

Acceptance is conditional on the manuscript's not being published elsewhere and on there being no announcement of this work to the newspapers, magazines, radio or television

until the publication date in Nature Structural & Molecular Biology.

Your paper will be published online soon after we receive proof corrections and will appear in print in the next available issue. You can find out your date of online publication by contacting the production team shortly after sending your proof corrections.

Please note that *Nature Structural & Molecular Biology* is a Transformative Journal (TJ). Authors may publish their research with us through the traditional subscription access route or make their paper immediately open access through payment of an article-processing charge (APC). Authors will not be required to make a final decision about access to their article until it has been accepted. Find out more about Transformative Journals

Sincerely,

Dimitris Typas
Associate Editor
Nature Structural & Molecular Biology
ORCID: 0000-0002-8737-1319